

# Dynamical correlation functions in the Ising field theory

**István Csépányi and Márton Kormos**

Department of Theoretical Physics, Institute of Physics,
Budapest University of Technology and Economics,
1111 Budapest, Műegyetem rkp. 3, Hungary
HUN-REN-BME Quantum Dynamics and Correlations Research Group,
Budapest University of Technology and Economics,
1111 Budapest, Műegyetem rkp. 3, Hungary
BME-MTA Statistical Field Theory 'Lendület' Research Group,
Budapest University of Technology and Economics,
1111 Budapest, Műegyetem rkp. 3, Hungary

## Abstract

We study finite temperature dynamical correlation functions of the magnetization operator in the one-dimensional Ising quantum field theory. Our approach is based on a finite temperature form factor series and on a Fredholm determinant representation of the correlators. While for space-like separations the Fredholm determinant can be efficiently evaluated numerically, for the time-like region it has convergence issues inherited from the form factor series. We develop a method to compute the correlation functions at time-like separations based on the analytic continuation of the space-time coordinates to complex values. Using this numerical technique, we explore all space-time and temperature regimes in both the ordered and disordered phases including short, large, and near-light-cone separations at low and high temperatures. We confirm the existing analytic predictions for the asymptotic behavior of the correlations except in the case of space-like correlations in the paramagnetic phase. For this case we derive a new closed form expression for the correlation length that has some unusual properties: it is a non-analytic function of both the space-time direction and the temperature, and its temperature dependence is non-monotonic.

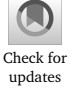

# 1 Introduction

The transverse field Ising model, or quantum Ising chain, is one of the most paradigmatic theories in quantum many-body physics [1,2]. It is a simple but insightful quantum mechanical system that serves as a toy model to study phase transitions and quantum magnetism. It can also be realized experimentally in compounds such as $CoNb_2O_6$ [3], $BaCo_2V_2O_8$ [4], $SrCo_2V_2O_8$ [5] as well as in quantum simulators employing trapped atoms [6].

By the quantum-classical mapping, it is related to the classical 2D Ising model. While the latter is the canonical example of continuous phase transitions, the quantum Ising chain is a paradigm of quantum criticality. Its importance stems, on the one hand, from its simplicity and, on the other hand, from the fact that it is exactly solvable by mapping it to a system of free fermions. However, this mapping is a nonlocal transformation, rendering the calculation of various physical quantities a highly nontrivial task. Crucially, the most important observable, the magnetization, can be expressed in terms of the fermions only in a complicated way, which explains why the model remains an actively researched area [7,8].

Here we revisit the dynamical (two-time) correlation functions of the magnetization in the quantum Ising chain, both at zero and finite temperatures. We focus on the vicinity of its quantum critical point where the correlation length is much larger than the lattice spacing and the physics is described, in the scaling limit, by the Ising quantum field theory.

As mentioned above, the nonlocal relation between the magnetization and the fermionic degrees of freedom poses profound challenges to calculating these correlators. There are various possible approaches to attack this problem. A simple and intuitive method, the so-called semiclassical approach was proposed in Ref. [9]. It is based on the idea that at low tempera-

tures, the excitations form a dilute gas and thus behave essentially classically. While it allows for the derivation of analytic expressions that give accurate results at low temperatures, it is not known how to turn it into a well-controlled, systematic method beyond the low-temperature regime.

Another approach is based on the knowledge of exact matrix elements of the magnetization in energy eigenstates, called form factors, that can be utilized in a spectral expansion. Unfortunately, the form factors are singular functions of the particle momenta, which makes the summation of the form factor series a notoriously hard problem [10–15]. Nevertheless, using the so-called representative state approach and exploiting the special structure of the Ising form factors, a partial resummation of the series was achieved in the recent Ref. [8], which allowed the authors to determine the asymptotic behavior of the dynamical correlation functions in the Ising spin chain. In the low-temperature limit, their results recover the semiclassical predictions. We note that the form factor expansion was also used to compute correlations in frequency and momentum space [15–18]. In a series of recent works, a method based on effective form factors was developed [19–21], but the dynamical correlators of the Ising model have not been studied yet.

There is another infinite series representation of the correlators in the Ising field theory based on "finite-temperature form factors" or, in a complementary formulation, on exact finite volume form factors [7, 22, 23]. In its original formulation, it can only be used for space-like separations (outside the light cone), but it is free of singularities and served as the basis of a few further analytic studies that focused on the asymptotic behavior [22, 24, 25].

Our starting point is the same finite temperature form factor series but we focus mainly on its numerical evaluation. Due to the special structure of the form factors, the correlation function can be reformulated as a Fredholm determinant [7, 26, 27]. This, in principle, opens the way to powerful techniques to determine its asymptotic behavior [28]. We leave this to the future and restrict ourselves to the numerical evaluation of the Fredholm determinant with the goal of exploring all the relevant regimes of the correlation function in terms of the ordered/disordered phase, the temperature, and the space-time separation. Despite the fact that the form factor series is well-behaved only for space-like separations, we develop a method to obtain numerical results also in the time-like region (inside the light cone).

Besides the numerical investigations, we also derive an apparently new analytical result for the asymptotic behavior of the correlator for large space-like separations in the paramagnetic phase. The result, which to our knowledge has not appeared in the literature, has some unusual features, e.g. a non-analytic and non-monotonic temperature dependence.

The paper is structured as follows. In Sec. 2, we review the fundamentals of the one-dimensional transverse Ising model and its scaling quantum field theory. In Sec. 3 we review the zero and finite temperature form factor series representation of the correlation functions. We also discuss the Fredholm determinant representation and how to evaluate it numerically, including our novel method in the time-like domain. Section 4 is devoted to the zero temperature correlation function, which serves as a benchmark for the numerical method. Section 5 contains our main results on the finite temperature correlations. We investigate the high temperature limit and study small and large separations in the space-like and time-like regions. We also compare our results to the existing theoretical predictions for the asymptotics and find agreement except in one case, where we present a new analytical expression. We give our summary and conclusions in Sec. 6. Some more involved derivations and technical details on the numerical method can be found in the appendices.

## 2 The quantum Ising chain and Ising field theory

The transverse field Ising model is a chain of interacting $S = 1/2$ spins arranged in a one-dimensional lattice. Each spin interacts with its nearest neighbors and is subject to a transverse magnetic field. The Hamiltonian describing this system captures the interplay of these interactions and is given by the following expression:

$$H = -J \sum_{j=1}^{N} \left( \hat{\sigma}_j^z \hat{\sigma}_{j+1}^z + g \hat{\sigma}_j^x \right). \tag{1}$$

In this equation, $J > 0$ sets the energy scale of the problem, $g$ is the relative strength of the external magnetic field compared to that of the nearest neighbor interaction, and $\hat{\sigma}_j^\alpha$ are the respective Pauli operators at site $j$. The index $j$ runs over the $N$ lattice sites of the chain and we impose periodic boundary conditions, $\hat{\sigma}_{N+1} = \hat{\sigma}_1$. The system has a quantum critical point at $g = 1$ that separates the two phases of the model. For $g < 1$, the $\hat{\sigma}^z \to -\hat{\sigma}^z$ symmetry is spontaneously broken and the system is in the ferromagnetically ordered phase with $\langle \hat{\sigma}^z \rangle \neq 0$. In the disordered or paramagnetic phase, the symmetry is restored and $\langle \hat{\sigma}^z \rangle = 0$.

The Hamiltonian (1) can be mapped to a system of free spinless fermions [29] (For an excellent and concise discussion, see Appendix A of Ref. [30]). The Jordan–Wigner transformation maps it to a quadratic expression in terms of the fermions which, employing a unitary transformation in the space of fermionic operators, can be brought to the canonical form

$$H^{R/NS} = \sum_{k_j \in R/NS} \epsilon(k_j) \hat{\gamma}_{k_j}^\dagger \hat{\gamma}_{k_j} + E_0^{R/NS}(N), \tag{2}$$

where NS and R refer to the sectors of the theory as discussed shortly. Here $\gamma_k$ are fermionic operators satisfying $\{\gamma_k^\dagger, \gamma_q\} = \delta_{k,q}$, and the dispersion relation is

$$\epsilon(k) = 2J \sqrt{1 - 2g \cos(k) + g^2}. \tag{3}$$

Note that at the critical point, $g = 1$, the dispersion relation $\epsilon(k) = 4J |\sin(k/2)|$ is gapless. By performing perturbative calculations around $g = 0$ and $g = \infty$, these particle-like excitations can be interpreted as domain walls between ordered domains of opposite magnetization in the ferromagnetic phase, and as spin flips in the transverse direction in the paramagnetic phase.

The Hilbert space splits into two sectors related to the boundary condition on the fermionic operators. In the Ramond (R) sector they have periodic, while in the Neveu–Schwarz (NS) sector they have antiperiodic boundary condition, which implies the quantization of momenta

$$k_j = \frac{2\pi}{N} j, \qquad j = -\frac{N}{2}, \dots, \frac{N}{2} - 1 \qquad \text{(Ramond)}, \tag{4a}$$

$$k_j = \frac{2\pi}{N} (j + 1/2), \qquad j = -\frac{N}{2}, \dots, \frac{N}{2} - 1 \qquad \text{(Neveu–Schwarz)}. \tag{4b}$$

It turns out that in the ferromagnetic phase, the number of fermions is always even,[1] while in the paramagnetic phase, states of an odd number of fermions are in the Ramond sector and states of an even number of fermions are in the Neveu–Schwarz sector. The Fock vacua of the two sectors have different energies denoted by $E_0^R(N)$ and $E_0^{NS}(N)$. For finite $N$, they satisfy $E_0^{NS}(N) < E_0^R(N)$, so the ground state is always in the NS sector. However, in the ferromagnetic phase, the energy difference of the vacua of the two sectors becomes exponentially small in $N$, signalling the spontaneous symmetry breaking occurring in the infinite system.

---

[1]This can be intuitively understood by the fact that the number of domain walls on a ring must be even.

We emphasize that the mapping from spins to fermions is nonlocal, so despite the quadratic Hamiltonian, the calculation of spin observables, e.g. their correlation functions, is a highly nontrivial task.

In the vicinity of the second order quantum phase transition, the correlation length becomes much larger than the lattice spacing, and the large scale behavior of the system can be described by a continuum theory. This scaling quantum field theory is the model that we study in this work. While there is no quantum phase transition at finite temperature, the field theory remains a good description near the critical point as long as the temperature is low compared to the lattice energy scale $J$. Due to universality, the scaling field theory describes all models in the Ising universality class such as spin chains having more complicated, e.g. next nearest neighbor, interactions that become irrelevant in the continuum limit.

The scaling limit is obtained by approaching the critical point, sending the lattice spacing to zero while sending the Ising interaction to infinity as

$$g \to 1, \ a \to 0, \ J \to \infty : \quad Na = L = \text{fixed}, \quad 2J|1-g| = \Delta = \text{fixed}, \quad 2Ja = c = \text{fixed}. \quad (5)$$

Here the length $L$ of the system, the energy gap $\Delta$, and the velocity $c$ is kept fixed. In this limit, the Hamiltonian (2) in the R and NS sectors becomes (with a slight abuse of notation)

$$H^{\text{R/NS}} = \sum_{p_j \in \text{R/NS}} \varepsilon(p_j)\hat{\eta}^\dagger_{p_j}\hat{\eta}_{p_j} + E_0^{\text{R/NS}}(L), \quad (6)$$

where $p_j = \hbar k_j/a$ are the momenta with spacing $\Delta p = 2\pi/L$, and the dispersion relation becomes the relativistic

$$\varepsilon(p) = \sqrt{\Delta^2 + p^2 c^2}, \quad (7)$$

which signals that the scaling field theory is relativistically invariant. Equation (7) shows that $c$ plays the role of the speed of light and the fermions have mass $m = \Delta/c^2$.

In what follows, we shall focus on the thermodynamic limit, $L \to \infty$, where summations like that in Eq. (6) become integrals. We will use extensively the relativistic rapidity $\theta$ in terms of which the momentum and energy are given by $p = mc\sinh\theta$ and $E = mc^2\cosh\theta$, respectively. From hereon, we set both the speed of light and Planck's constant to 1, i.e. we work in units where $c = 1, \hbar = 1$.

## 3 Dynamical correlation functions of the magnetization

We will be concerned with the equilibrium correlation function of the magnetization operator $\hat{\sigma}(x,t)$ in the field theory. In our normalization, it is related to $\hat{\sigma}^z_j$ on the lattice by

$$\hat{\sigma}(x = ja) = (2a)^{-1/8}\hat{\sigma}^z_j, \quad (8)$$

so its expectation value in the ferromagnetic ground state is given by $\langle\hat{\sigma}(x)\rangle = m^{1/8}$.

The dynamical correlation function of the magnetization at finite temperature is

$$C(x,t;\beta) = \langle\hat{\sigma}(x,t)\hat{\sigma}(0,0)\rangle_\beta = \frac{\text{Tr}\left\{e^{-\beta\hat{H}}\hat{\sigma}(x,t)\hat{\sigma}(0,0)\right\}}{\text{Tr}\,e^{-\beta\hat{H}}}, \quad (9)$$

where $\beta$ is the inverse temperature and the operators are in the Heisenberg picture,

$$\hat{\sigma}^z(x,t) = e^{iHt}e^{-iPx}\hat{\sigma}^z(0,0)e^{iPx}e^{-iHt}. \quad (10)$$

At zero temperature,

$$C(x,t) = \langle 0|\hat{\sigma}^z(x,t)\hat{\sigma}^z(0,0)|0\rangle, \quad (11)$$

where $|0\rangle$ denotes the ground state. With our normalization, the short distance singularity of the correlators is given by [31]

$$\langle \hat{\sigma}(x,0)\hat{\sigma}(0,0)\rangle_\beta \underset{x\to 0}{\sim} \frac{2^{-1/6}e^{1/4}\mathcal{A}^{-3}}{x^{1/4}}, \tag{12}$$

where $\mathcal{A} = 1.2824271291\ldots$ is Glaisher's constant.

Let us note that in the space-like region, the correlators are real functions. Indeed, $\langle \hat{O}_1\hat{O}_2\rangle^* = \langle \hat{O}_2^\dagger\hat{O}_1^\dagger\rangle$, so if the two operators are hermitian and commute with each other then $\langle \hat{O}_1\hat{O}_2\rangle^* = \langle \hat{O}_1\hat{O}_2\rangle$. In a relativistic field theory, two operators with space-like separations must commute due to causality, so $C(x,t;\beta)^* = C(x,t;\beta)$ for $x^2 > t^2$. Therefore, the imaginary parts of the correlators are non-trivial only for time-like separations.

By dimensional arguments, the correlation function must have the form

$$C_{\text{f/p}}(x,t;\beta) = m^{1/4}\tilde{C}_{\text{f/p}}(mx,mt;m\beta), \tag{13}$$

where $\tilde{C}_{\text{f/p}}$ is a dimensionless function and the subscript refers to the ferromagnetic and paramagnetic phase.

## 3.1 Form factor expansion

A possible approach to evaluate the correlation functions is using a spectral expansion obtained by inserting a resolution of identity in terms of energy and momentum eigenstates between the operators and expanding the thermal trace in the same basis. In the Ising field theory, these eigenstates are the multiparticle states of free fermions, $|\theta_1,\ldots,\theta_k\rangle$, characterized by the rapidities of particles that satisfy

$$\hat{H}|\theta_1,\ldots,\theta_k\rangle = \left(\sum_{i=1}^k m\cosh(\theta_i)\right)|\theta_1,\ldots,\theta_k\rangle,$$
$$\hat{P}|\theta_1,\ldots,\theta_k\rangle = \left(\sum_{i=1}^k m\sinh(\theta_i)\right)|\theta_1,\ldots,\theta_k\rangle. \tag{14}$$

In infinite volume, the rapidities are continuous and the corresponding resolution of identity is given by

$$\mathbb{I} = \sum_{k=0}^\infty \frac{1}{k!}\int_{-\infty}^\infty \frac{d\theta_1}{2\pi}\cdots\int_{-\infty}^\infty \frac{d\theta_k}{2\pi}|\theta_1,\ldots,\theta_k\rangle\langle\theta_1,\ldots,\theta_k|. \tag{15}$$

The operator matrix elements between these multiparticle states are called form factors, and the spectral representation in terms of multiparticle states (here and in integrable models in general) is usually referred to as the form factor expansion.

### 3.1.1 Zero temperature

At zero temperature, the correlation function is evaluated in the ground state so the form factor expansion is given by

$$C(x,t) = \sum_{k=0}^\infty \frac{1}{k!}\prod_{i=0}^k\left(\int_{-\infty}^\infty \frac{d\theta_i}{2\pi}e^{imx\sinh\theta_i - imt\cosh\theta_i}\right)|\langle\theta_1,\ldots,\theta_k|\hat{\sigma}(0,0)|0\rangle|^2. \tag{16}$$

The matrix elements of the magnetization operator are known explicitly. The only nonzero matrix elements of the $\hat{\sigma}$ operator are between states belonging to different sectors. The form factors between the ground state and the multiparticle states in an infinite system are given by [32]

$$\langle \theta_1, \dots, \theta_k | \hat{\sigma} | 0 \rangle = i^{[n/2]} m^{1/8} \prod_{1 \le i < j \le k} \tanh\left(\frac{\theta_i - \theta_j}{2}\right), \tag{17}$$

where $k$ must be even in the ferromagnetic phase and odd in the paramagnetic phase, otherwise the matrix elements are zero. In conclusion, the zero temperature correlation functions of the magnetization can be expressed as

$$C(x, t; 0) = m^{1/4} \sum_{N=0}^{\infty}{}' \frac{1}{N!} \prod_{j=0}^{N} \left( \int_{-\infty}^{\infty} \frac{d\theta_j}{2\pi} e^{imx \sinh \theta_j - imt \cosh \theta_j} \right) \prod_{1 \le k < l \le N} \tanh^2\left(\frac{\theta_k - \theta_l}{2}\right), \tag{18}$$

where the primed sum runs over nonnegative *even* integers in the *ferromagnetic* phase and over positive *odd* integers in the *paramagnetic* phase.

In the space-like region, $x^2 > t^2$, the series (18) is a convergent large distance expansion. This can be seen by shifting the integration contour parallel to the real axis to the $\text{Im}(\tilde{\theta}) = \pi/2$ line where the exponents become

$$imx \sinh(\theta + i\pi/2) - imt \cosh(\theta + i\pi/2) = -mx \cosh \theta + mt \sinh \theta = -m\sqrt{x^2 - t^2} \cosh(\theta - \theta_0), \tag{19}$$

where $\tanh \theta_0 = t/x$. This shows that the $N$th term decays as $\sim e^{-Nm\sqrt{x^2 - t^2}}$. The truncated form factor series was numerically evaluated in [31].

In the time-like region, $t^2 > x^2$, a different contour deformation can be used to make the integrands decay at large rapidities. The sign of the real and imaginary parts of the rapidity must be the same for large rapidities, which can be achieved by a contour deformation $\theta \to \theta + is(\theta)$ leading to

$$imx \sinh(\theta + is(\theta)) - imt \cosh(\theta + is(\theta))$$
$$= m\sqrt{t^2 - x^2}\left[-i \cos s(\theta) \cosh(\theta - \theta_0') - \sin s(\theta) \sinh(\theta - \theta_0')\right], \tag{20}$$

where now $\tanh \theta_0' = x/t$. It follows that as $\theta \to \pm\infty$, $\text{sign}(s(\theta)) = \text{sign}(\theta)$ should hold: the optimal choice is a regularized version of $s(\theta) = \pi/2 \, \text{sign}(\theta - \theta_0')$, e.g. $\pi/2 \tanh[\alpha(\theta - \theta_0')]$. Note that the real part can be arbitrarily small, so we cannot make a claim about the exponential suppression of multiparticle terms similar to the space-like case.

### 3.1.2 Finite temperature

The finite temperature generalization of the form factor expansion is a notoriously difficult task [14, 15]. The reason is that due to the thermal trace, it becomes a double sum that contains multi-particle matrix elements like [32]

$$\langle \theta_1, \dots \theta_n | \hat{\sigma} | \theta_1', \dots \theta_m' \rangle = i^{[(n+m)/2]} m^{1/8} \frac{\displaystyle\prod_{1 \le i < j \le n} \tanh\left(\frac{\theta_i - \theta_j}{2}\right) \prod_{1 \le k < l \le m} \tanh\left(\frac{\theta_k' - \theta_l'}{2}\right)}{\displaystyle\prod_{i=1}^{n} \prod_{k=1}^{m} \tanh\left(\frac{\theta_i - \theta_k'}{2}\right)}. \tag{21}$$

Unfortunately, these form factors have singularities, so-called kinematic poles, when a rapidity in the bra state is equal to another in the ket state. Since all the rapidities are integrated over, this implies that the naive finite temperature form factor expansion is divergent, and a more

careful analysis is needed to obtain meaningful results [11–13]. A possible approach is to use finite volume regularization [14, 15, 33, 34], but it has to be worked out for each term separately, so the (partial) resummation of the series remains a difficult task.

However, in the Ising field theory, a well-defined form factor series was derived in Refs. [7, 22]. It was obtained by using finite temperature form factors and also by analytically continuing a Euclidean correlation function computed by swapping the roles of space and Euclidean time. The original correlation function is defined on an infinite cylinder whose circumference corresponds to the inverse temperature. Swapping space and time yields a finite system with periodic boundary condition, at zero temperature. In this picture, the correlation function can be computed by a zero-temperature form factor expansion, provided that the finite volume form factors are available. The Ising model is, to our knowledge, the unique system where these are known exactly. For the quantum Ising chain, they were extracted from calculations in the classical 2D Ising model [35–37]. The corresponding Ising field theory form factors follow from the scaling limit, but they were also derived independently within the field theory [38].

For the sake of completeness, we present the derivation based on finite volume form factors in Appendix A. The result is the form factor series for the finite temperature two-point function,

$$C(x, t; \beta) = m^{1/4} S(m\beta)^2 e^{-\Delta \mathcal{E}(\beta)x} \tag{22}$$

$$\times \sum_{N}^{\infty}{}' \frac{1}{N!} \sum_{\epsilon_1, \dots, \epsilon_N = \pm} \prod_{j=0}^{N} \int_{-\infty+i\epsilon_j\delta}^{\infty+i\epsilon_j\delta} \frac{d\theta_j}{2\pi} \frac{e^{im\epsilon_j(x\sinh\theta_j - t\cosh\theta_j) + \epsilon_j\eta(\theta)}}{\epsilon_j \left(1 - e^{-\epsilon_j m\beta \cosh\theta_j}\right)} \prod_{1 \le k < l \le N} \tanh\left(\frac{\theta_k - \theta_l}{2}\right)^{2\epsilon_i\epsilon_j},$$

where, just like in the $T = 0$ case in Eq. (18), the first primed sum runs over nonnegative *even* integers in the *ferromagnetic* phase and over positive *odd* integers in the *paramagnetic* phase. The temperature-dependent factors are given by the integrals

$$S(m\beta) = \exp\left[\frac{(m\beta)^2}{2} \int_{-\infty}^{\infty} \frac{d\theta_1 d\theta_2}{(2\pi)^2} \frac{\sinh\theta_1 \sinh\theta_2}{\sinh(m\beta\cosh\theta_1)\sinh(m\beta\cosh\theta_2)} \log\left|\coth\left(\frac{\theta_1 - \theta_2}{2}\right)\right|\right], \tag{23}$$

$$\Delta\mathcal{E}(\beta) = \int_{-\infty}^{\infty} \frac{d\theta}{2\pi} m\cosh\theta \log\left[\frac{1 + e^{-m\beta\cosh\theta}}{1 - e^{-m\beta\cosh\theta}}\right], \tag{24}$$

and

$$\eta(\theta) = \int_{-\infty}^{\infty} \frac{d\theta'}{\pi i} \frac{1}{\sinh(\theta - \theta')} \log\left[\frac{1 + e^{-m\beta\cosh\theta'}}{1 - e^{-m\beta\cosh\theta'}}\right]. \tag{25}$$

Finally, $\delta > 0$ is a small positive constant governing the contour prescription.

The finite temperature series (22) is much more complicated than its zero temperature counterpart (18). The temperature-dependent statistical factors in the denominator and the summation over $\epsilon_j = \pm$, which can be interpreted as summing over particle and hole excitations, make it similar to the Leclair–Mussardo proposal [11], but there are important differences. First, the contour prescription avoids the kinematic poles of the form factors present when $\epsilon_i = -\epsilon_j$ and makes the expression well-defined and finite. Note that the direction of the contour shifts depends on $\epsilon$, that is, it is the opposite for particles and holes. Second, there are additional temperature-dependent factors (23), (24), (25) beyond the thermal statistical factors that are missing from the Leclair–Mussardo formula. The $e^{-\Delta \mathcal{E} x}$ term gives a qualitatively important feature: it encodes an exponential decay in $x$ having a purely thermal origin. Finally, the statistical factors are not of Fermi–Dirac but of Bose–Einstein type, which may be related to the semilocality properties of the magnetization and the fermion creating operators. To derive these features from a naive low-temperature form factor expansion would require a partial resummation of infinitely many terms.

Equation (22) has one vital restriction: in its present form, it can only be used for spatial separations, i.e. in the $|x| > |t|$ region. Here the space and time-dependent exponentials can be made decay for large rapidities by increasing the contour shifts up to $\delta = \pi/2 - \delta'$ with $\delta' > 0$ being a positive number necessary to avoid the poles of the thermal denominator. This follows from an analysis similar to Eq. (19) and implies that the terms of the series are exponentially suppressed in the number of particles, so it is a large-distance expansion.

For time-like separations, each term in the series is divergent because the contour prescription makes the space-time exponentials blow up for large rapidities. To make the integrals convergent, we would need to deform the contours similarly to Eq. (20). But this is nontrivial because then the contours would have to pass the kinematical poles of the form factors when the rapidity of a particle and a hole coincide. To solve this issue, we would have to sum up the resulting pole contributions as it was suggested in Ref. [7]. We discuss a workaround to this problem below in Sec. 3.2.1.

## 3.2 Fredholm determinant representation

The infinite form factor series (22) can be recast in terms of a Fredholm determinant [7, 26]. We provide an outline of this procedure below with additional details available in Appendix B.

The key observation is that the double product can be expressed as a determinant. Introducing the $u_i = e^{\theta_i}$ variables,

$$
\begin{aligned}
\prod_{1 \leq i < j \leq k} \tanh\left(\frac{\theta_i - \theta_j}{2}\right)^{2\epsilon_i \epsilon_j} &= \prod_{1 \leq i < j \leq k} \left(\frac{u_i - u_j}{u_i + u_j}\right)^{2\epsilon_i \epsilon_j} \\
&= \prod_{1 \leq i < j \leq k} \left(\frac{\epsilon_i u_i - \epsilon_j u_j}{\epsilon_i u_i + \epsilon_j u_j}\right)^2 = \det\left[\frac{2\epsilon_i u_i}{\epsilon_i u_i + \epsilon_j u_j}\right]_{i,j=1}^{k},
\end{aligned}
\tag{26}
$$

where the second equality can be checked to hold for all four $\epsilon_{i,j} = \pm 1$ combinations, and the third one is the result of the Cauchy identity

$$
\det\left(\frac{1}{x_i + y_j}\right)_{i,j=1}^{k} = \frac{\displaystyle\prod_{1 \leq i < j \leq k} (x_j - x_i)(y_j - y_i)}{\displaystyle\prod_{1 \leq i,j \leq k} (x_i + y_j)},
\tag{27}
$$

applied in the special case $x_i = y_i = \epsilon_i u_i$. Upon substituting Eq. (26) into Eq. (22), the combination of ferromagnetic and paramagnetic correlators become

$$
C_{\pm}(x, t; \beta) \equiv C_{\mathrm{f}}(x, t; \beta) \pm C_{\mathrm{p}}(x, t; \beta) = m^{1/4} S(m\beta) e^{-\Delta \mathcal{E} x} \mathrm{Det}\left(\mathbb{I} + \tilde{\mathbf{K}}_{x,t;\beta}^{\pm}\right),
\tag{28}
$$

where the Fredholm determinant is defined as

$$
\mathrm{Det}\left(\mathbb{I} + \tilde{\mathbf{K}}_{x,t;\beta}^{\pm}\right) = \sum_{N=0}^{\infty} \sum_{\epsilon_1,\dots\epsilon_N = \pm} \int_{-\infty}^{\infty} \frac{d\theta_1 \dots d\theta_N}{N!} \det\left[K_{\epsilon_i, \epsilon_j | x,t;\beta}^{\pm}(\theta_i, \theta_j)\right]_{i,j=1}^{N},
\tag{29}
$$

with the kernel

$$
K_{\epsilon, \epsilon'|x,t;\beta}^{\pm}(\theta, \theta') = \frac{\pm e^{im\epsilon(x \sinh(\theta + i\epsilon\delta) - t\cosh(\theta + i\epsilon\delta)) + \epsilon\eta(\theta + i\epsilon\delta)}}{2\pi\epsilon(1 - e^{-\epsilon m\beta \cosh(\theta + i\epsilon\delta)})}\left(\frac{2\epsilon \, e^{\theta + i\epsilon\delta}}{\epsilon \, e^{\theta + i\epsilon\delta} + \epsilon' \, e^{\theta' + i\epsilon'\delta}}\right).
\tag{30}
$$

In the space-like region, this Fredholm determinant can be numerically evaluated in the following way [39]. Utilizing the contour shifts with $0 < \delta < \pi/2$, the integrand decays exponentially as $|\theta| \to \infty$, so the rapidity integrals can be restricted to a finite $[-\vartheta, \vartheta]$ interval.

Discretizing this interval by dividing it into $n$ equal pieces of length $\Delta\theta = 2\vartheta/n$ gives the discrete set of rapidities $\{\theta_a\} = \{\theta_1 = -\vartheta, \ldots, \theta_n = \vartheta - \Delta\theta\}$. Then (suppressing some variables for the ease of notation) the Fredholm determinant $\text{Det}(\mathbb{I} + \mathbf{K})$ can be approximated by the determinant of a $2n \times 2n$ matrix,

$$D(n) = \det[\mathbb{I} + \Delta\theta \tilde{K}(n)], \tag{31}$$

where

$$\tilde{K}(n) = \begin{pmatrix} \{K_{++}(\theta_a, \theta_b)\} & \{K_{+-}(\theta_a, \theta_b)\} \\ \{K_{-+}(\theta_a, \theta_b)\} & \{K_{--}(\theta_a, \theta_b)\} \end{pmatrix}_{2n \times 2n}. \tag{32}$$

In the limit $n \to \infty$, the finite determinant approaches the Fredholm determinant, $D(n) \to \text{Det}(\mathbb{I} + \mathbf{K})$.

The zero temperature series can also be formulated as a Fredholm determinant (see also Appendix B). This simpler Fredholm determinant can be formally obtained from the finite temperature one by implementing the contour prescriptions (19) for space-like and (20) for time-like separations through $\delta(\theta)$, and fixing all $\epsilon_j$ to $+1$, i.e. dropping the discrete sum in Eq. (29) and restricting the $\tilde{K}$ matrix to its $K_{++}$ block involving the kernel

$$K_{x,t}^{\pm}(\theta, \theta') = \frac{\pm 1}{2\pi} e^{im\epsilon(x \sinh(\theta + i\epsilon\delta) - t \cosh(\theta + i\epsilon\delta))} \left( \frac{2 e^{\theta + i\delta}}{e^{\theta + i\delta} + e^{\theta' + i\delta}} \right). \tag{33}$$

### 3.2.1 Numerical method in the time-like region

As we mentioned already, applying the method to the finite temperature time-like region is infeasible, because for $|x| < |t|$, the exponential in the numerator of the kernel (30) blows up for large rapidities. This means that each term in the form factor series (22) is ill-defined, and they may be possible to convert into well-defined expressions only by a nontrivial contour deformation [7].

Surprisingly, there is an alternative route that makes sense of the series and allows us to extract finite numerical results from the Fredholm determinant. The idea is to keep the contours of Eq. (22) intact while analytically continuing the $\zeta = t/x$ value to the complex plane. Then the calculations can be carried out in the time-like region, $\text{Re}\,\zeta > 1$, for sufficiently small imaginary parts and the physical results are obtained by extrapolating the results back to the real line. However, the exponentials for particles and holes require opposite $\text{Im}\,\zeta$, so the series must be generalized by writing

$$C(x, t; \beta) \to C(x, \zeta_+, \zeta_-; \beta), \tag{34}$$

where $\zeta_+$ and $\zeta_-$ are the complex parameters used in the particle ($\epsilon = +$) and hole ($\epsilon = -$) terms:

$$C(x, \zeta_+, \zeta_-; \beta) = m^{1/4} S(m\beta)^2 e^{-\Delta\mathcal{E}(\beta)x} \tag{35}$$

$$\times \sum_{N}{}' \frac{1}{N!} \sum_{\epsilon_1, \ldots, \epsilon_N = \pm} \prod_{j=0}^{N} \int_{-\infty + i\epsilon_j\delta}^{\infty + i\epsilon_j\delta} \frac{d\theta_j}{2\pi} \frac{e^{imx\epsilon_j(\sinh\theta_j - \zeta_{\epsilon_j} \cosh\theta_j) + \epsilon_j\eta(\theta)}}{\epsilon_j \left(1 - e^{-\epsilon_j m\beta \cosh\theta_j}\right)} \prod_{1 \le k < l \le N} \tanh\left( \frac{\theta_k - \theta_l}{2} \right)^{2\epsilon_i\epsilon_j}.$$

The physical correlators correspond to the $\text{Re}\,\zeta_+ = \text{Re}\,\zeta_-$ and $\text{Im}\,\zeta_+ = \text{Im}\,\zeta_- = 0$ values. The strategy is to perform simulations sufficiently close to the desired physical point and then extrapolate the results to obtain the physical values.

It is a simple exercise to check that in the $\text{Re}\,\zeta_\pm > 1$ time-like region, the condition for decaying exponentials is

$$\text{Im}\,\zeta_+ < -\tan\delta\,(\text{Re}\,\zeta_+ - 1), \tag{36a}$$

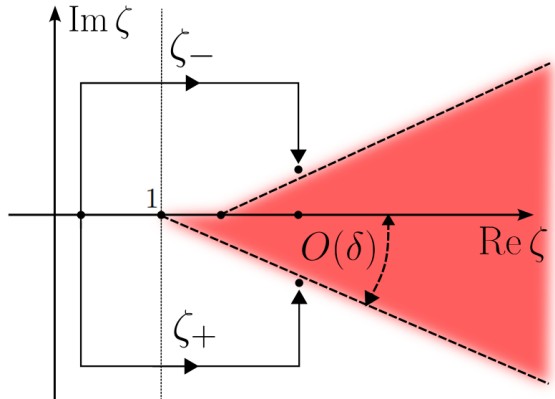

Figure 1: Sketch of the method used in the finite temperature time-like domain. The parameter $\zeta = t/x$ is analytically continued to the complex plane separately for the particles ($\zeta_+$) and the holes ($\zeta_-$). The physical points corresponds to $\mathrm{Re}\,\zeta_+ = \mathrm{Re}\,\zeta_-$ and $\mathrm{Im}\,\zeta_+ = \mathrm{Im}\,\zeta_- = 0$. The red wedge-like region corresponds to the domain where the integrands blow up. The size of the forbidden sector is proportional to $\delta$, the imaginary part of the rapidities.

for particles and

$$\mathrm{Im}\,\zeta_- > \tan\delta\,(\mathrm{Re}\,\zeta_- - 1) - \beta/x \,, \tag{36b}$$

for holes. Note that here the temperature-dependent denominator also contributes to the asymptotic behavior, leading to a less restrictive criterion for the imaginary value of $\zeta_-$.

Figure 1 provides a schematic drawing of the method described above. Equations (36) imply that there is a forbidden region in the complex $\zeta$ plain where certain terms in Eq. (35) diverge. It is also clear that the size of this region is proportional to $\delta$, the complex rapidity shift. Therefore in order to perform calculations sufficiently close to a physical point, a respective small $\delta$ must be chosen. Performing several of these calculations can, in principle, enable us to extrapolate our results to the desired physical points.

There are, however, two caveats that concern the validity of this procedure; both of them are linked to the size restriction on the $\delta$ rapidity shift. First, if $\delta$ is small, the relative rapidity values between particles and holes can become small, which reinstates the problem of kinematic poles. Indeed, for such parameter configurations, the Fredholm determinant kernels display sharp peaks concentrated on $O(\delta)$ rapidity domains, which means that our discretization has to be fine enough to incorporate this new feature. Second, when the complex rapidity shift is tiny and the $\zeta_\pm$ parameters are close to their respective boundaries in Figure 1, the exponential contributions in Eq. (35) give rise to rapid oscillations. This gives another requirement on the necessary rapidity discretization. In practice, both of these difficulties could be overcome by using a less rigid contour prescription for the rapidities as long as the physical $\zeta$ value is not too large. It turns out that the necessary matrix size needed for the calculations is proportional to $\zeta - 1$. Appendix C provides some details on this procedure.

## 4 Numerical simulations at zero temperature

Even though our main focus is the finite temperature correlations, we start by discussing the zero temperature case. Matching the well-known analytical results in the short and large distance limits provides a proof of principle for the numerical evaluation of the form factor series via calculating Fredholm determinants.

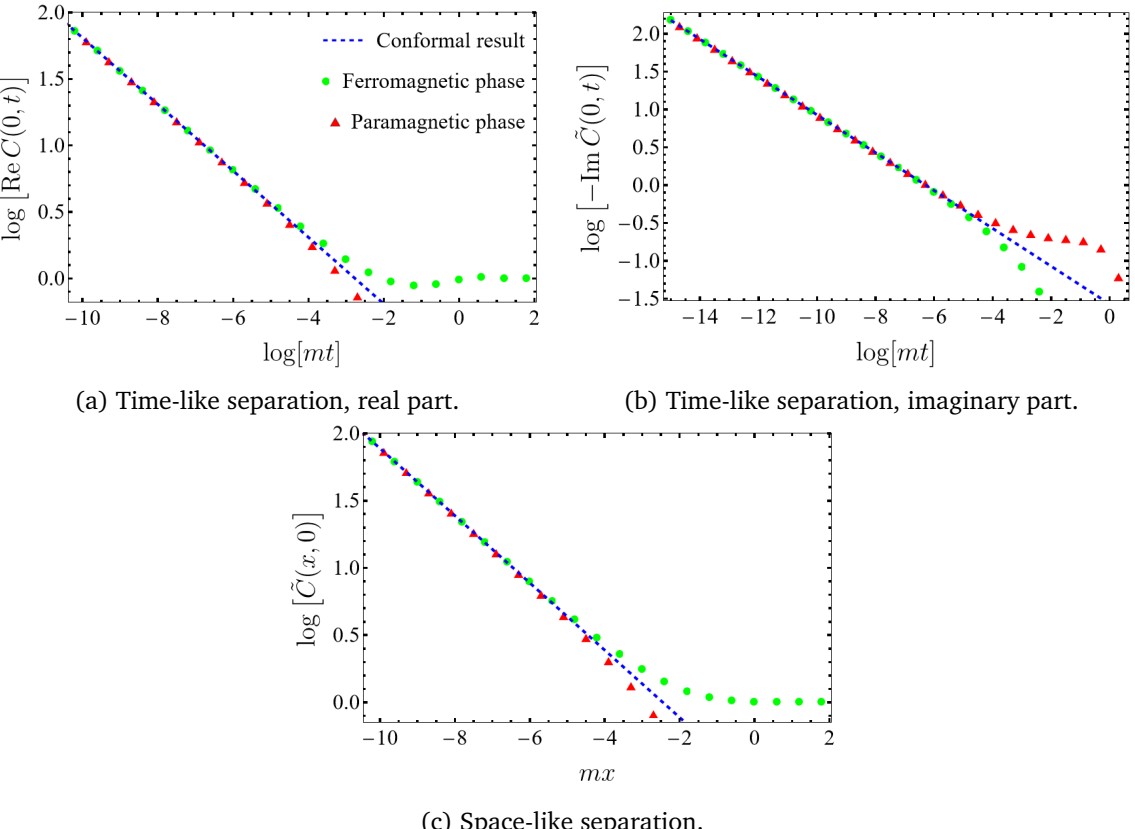

(a) Time-like separation, real part.

(b) Time-like separation, imaginary part.

(c) Space-like separation.

Figure 2: Zero temperature correlation functions for small separations. In the time-like domain, the real and imaginary parts are shown in separate plots. In all three cases, both the paramagnetic and ferromagnetic curves approach the analytic results (38) shown as green dashed lines. The dimensionless $\tilde{C}$ is defined in Eq. (13).

At zero temperature, the correlation functions are Lorentz invariant, i.e. they only depend on the relativistic interval, so $C(x,t) = C(\sqrt{x^2-t^2},0)$ for space-like and $C(x,t) = C(0,\sqrt{t^2-x^2})$ for time-like separations. This can be seen explicitly from Eqs. (19) and (20). Therefore it is enough to consider the equal time (static) correlator $C(x,0)$ and the autocorrelation function $C(0,t)$. Below we investigate these functions in both phases of the model.

## 4.1 Near the light cone

For small separations, $mx \ll 1$, the equal-time correlation function is unaffected by the mass and as $mx \to 0$, we should recover the conformal field theory correlator (12) in both the paramagnetic and ferromagnetic phases. By Lorentz invariance, it implies that for $0 < m^2(x^2-t^2) \ll 1$, the correlation function should behave as[2]

$$C_{\text{f/p}}(x,t) \sim \frac{2^{-1/6}e^{1/4}\mathcal{A}^{-3}}{(x^2-t^2)^{1/8}}. \tag{37}$$

---

[2]Note that this is a property of the field theory and does not imply a similar short distance behavior in the spin chain.

This expression can be viewed as the conformal result, depending on $x^2 + \tau^2$, analytically continued to real time as $\tau = it$. It is not obvious that the same analytic continuation works also in the time-like regime but we will see shortly that the formula (37) applies in the time-like case, too. The equal-time and equal-space correlation functions thus behave for small separations as

$$C_{f/p}(x,0) \xrightarrow{x\to 0} 2^{-1/6}e^{1/4}\mathcal{A}^{-3}x^{-1/4}\,, \tag{38a}$$

$$C_{f/p}(0,t) \xrightarrow{t\to 0} 2^{-1/6}e^{1/4}\mathcal{A}^{-3}e^{-i\pi/8}t^{-1/4}\,. \tag{38b}$$

We computed these correlators by numerically evaluating the Fredholm determinant, as discussed in Sec. 3.2. The results are plotted in Fig. 2 together with the predictions in Eqs. (38). We find that both the ferromagnetic and paramagnetic correlators approach the conformal power laws.

For the equal time correlator, similar numerical results can be obtained by computing the first few terms of the form factor series (18), e.g. by using a Monte Carlo integration technique as was done in Ref. [31]. However, to achieve the precision of our numerical results, the contribution of several terms must be taken into account.

## 4.2 Large separations

As mentioned earlier, for space-like separations, the form factor expansions are large-distance expansions in the sense that the $N$-particle terms in the series are suppressed as $e^{-Nm\sqrt{x^2-t^2}}$, so we can extract their asymptotic behavior as $m\sqrt{x^2-t^2} \to \infty$ by calculating the first non-trivial terms in Eq. (18). Although such a general statement is not true for time-like separations, below we provide numerical evidence that this is indeed the case.

In the paramagnetic phase, the first term is a one-dimensional integral that can be evaluated analytically using the identity

$$\int_{-\infty}^{\infty} \frac{d\theta}{2\pi} e^{imx\sinh\theta - imt\cosh\theta} = \frac{1}{\pi}K_0\left(m\sqrt{x^2-t^2}\right)\,, \tag{39}$$

where $K_0$ is the zeroth modified Bessel function of the second kind. Therefore, the large-separation behavior of the paramagnetic correlators is given by

$$C_p(x,0) \approx \frac{1}{\pi}K_0\left(m\sqrt{x^2-t^2}\right) \approx \frac{m^{1/4}}{\sqrt{2\pi}}\frac{e^{-m\sqrt{x^2-t^2}}}{\sqrt{m}\,(x^2-t^2)^{1/4}}\,. \tag{40}$$

In particular, for the equal-space and equal-time correlators,

$$C_p(x,0) \xrightarrow{x\to\infty} \frac{m^{1/4}}{\sqrt{2\pi}}\frac{e^{-mx}}{\sqrt{mx}}\,, \tag{41a}$$

$$C_p(0,t) \xrightarrow{t\to\infty} \frac{m^{1/4}}{\sqrt{2\pi}}\frac{e^{-imt-i\pi/4}}{\sqrt{mt}}\,. \tag{41b}$$

In the ferromagnetic phase, the first nontrivial term is the two-dimensional integral

$$\frac{1}{2}\int_{-\infty}^{\infty} \frac{d\theta_1 d\theta_2}{(2\pi)^2} e^{imx(\sinh\theta_1+\sinh\theta_2)-imt(\cosh\theta_1+\cosh\theta_2)}\tanh^2\left(\frac{\theta_1-\theta_2}{2}\right)\,. \tag{42}$$

Upon introducing the $u = \theta_1 + \theta_2$ and $v = \frac{1}{2}(\theta_1 - \theta_2)$ variables, we can carry out the $u$ integral using Eq. (39), which yields the asymptotics

$$m^{-1/4}C_f(x,0) \xrightarrow{x\to\infty} 1 + \frac{1}{2\pi^2}\int_{-\infty}^{\infty} dv K_0\left(2m\sqrt{x^2-t^2}\cosh v\right)\tanh^2(v) \approx 1 + \frac{e^{-2m\sqrt{x^2-t^2}}}{8\pi m^2(x^2-t^2)}\,. \tag{43}$$

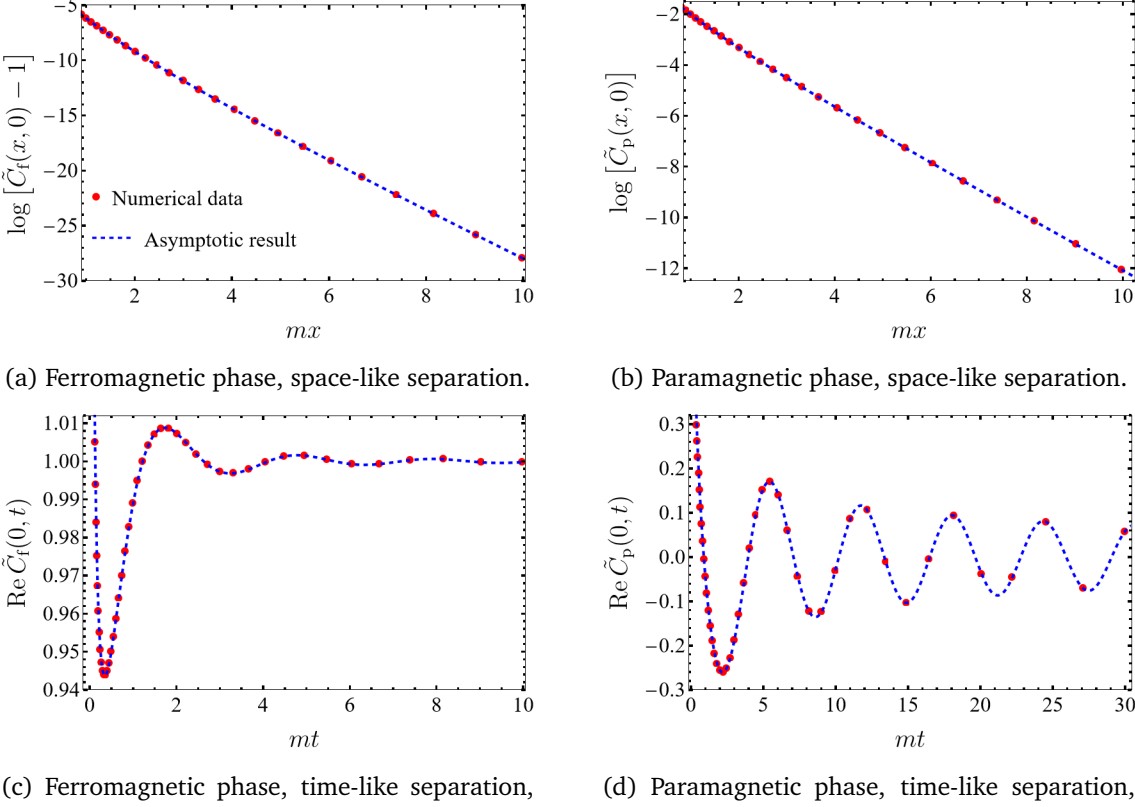

(a) Ferromagnetic phase, space-like separation.

(b) Paramagnetic phase, space-like separation.

(c) Ferromagnetic phase, time-like separation, real part.

(d) Paramagnetic phase, time-like separation, real part.

Figure 3: Asymptotic behavior of the zero temperature correlation functions. The left (right) column corresponds to the ferromagnetic (paramagnetic) phases. Our numerical results are shown in red dots, the blue dashed lines represent the first terms of the form factor series, Eq. (40) and the integral in Eq. (43).

For time-like separations, $\sqrt{x^2 - t^2}$ is replaced by $i\sqrt{t^2 - x^2}$.

For the $x = 0$ autocorrelation functions, the explicit expressions in Eqs. (41) and (43) agree with the leading terms in the scaling limit of the asymptotic expansion of the lattice autocorrelation function computed in Ref. [40].

We compare the asymptotic predictions (the Bessel function in (40) and the integral in (43)) with the numerical evaluation of the Fredholm derminant in Fig. 3. In all cases, the results of the simulations follow the asymptotic expressions remarkably well. For time-like separations, we only plot the real parts but we checked that the imaginary parts show equally good agreement. This shows that the first terms in the form factor expansion give the asymptotic behavior also for time-like separations.

## 5 Finite temperature results

After discussing the zero temperature correlation functions, we turn to the main subject of our work, the finite temperature dynamical correlation functions. These correlators are not Lorentz invariant anymore, which, together with the additional temperature dependence, implies a much richer set of characteristics. We first study them in the limit of small separations and then study their high temperature features. Then we turn to the investigation of their asymptotic behavior in the limit of large separations. In all these cases we study both the ferromagnetic and paramagnetic correlators.

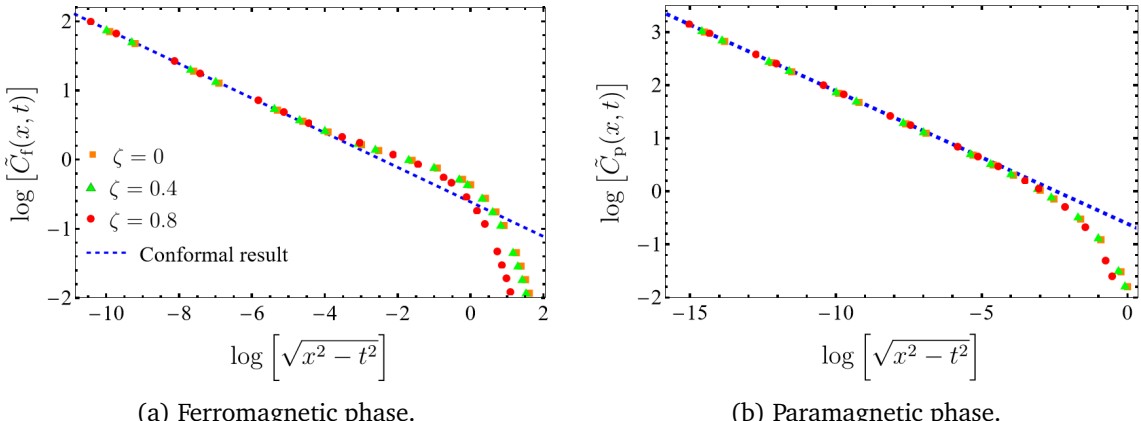

(a) Ferromagnetic phase.

(b) Paramagnetic phase.

Figure 4: Dynamical correlation functions at inverse temperature $m\beta = 1$ for different $\zeta = t/x$ rays plotted against the Lorentz invariant separation. The conformal expression (37) is shown in a blue dashed line.

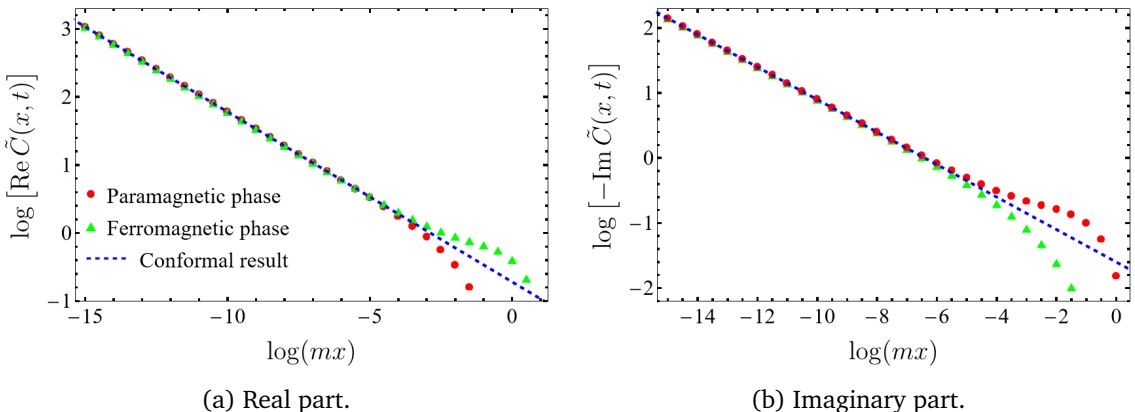

(a) Real part.

(b) Imaginary part.

Figure 5: Finite temperature time-like correlation functions for small separations performed at $\zeta = 1.5$ and $m\beta = 1$. The conformal expression (37) is shown in a blue dashed line.

## 5.1 Small separations

If the separation of the two operators is smaller than the two inherent length scales of the theory, $1/m$ and $\beta$, the effects of both the mass gap and the temperature become negligible. As a consequence, the correlators are governed by the zero temperature conformal result (37) which is also Lorentz invariant.

We check this behavior in Fig. 4 by plotting multiple dynamical correlation functions at inverse temperature $m\beta = 1$ for different space-like $\zeta = t/x < 1$ rays against the Lorentz invariant separation. The plots confirm that all the curves follow the same asymptotic behavior given by (37) for small separations. For larger distances, Lorentz invariance breaks down, as expected, and the space-time dependence of the correlators becomes more complicated.

Using our time-like algorithm described in Sec. 3.2.1, we also study the time-like domain where the form factor series in its form in Eq. (22) breaks down. We show our results for the real and imaginary part of the correlation functions at $\zeta = 1.5$, obtained by the extrapolation procedure of Sec. 3.2.1, in Fig. 5. The plots demonstrate that the correlators obey the conformal law for small separations even in the time-like case.

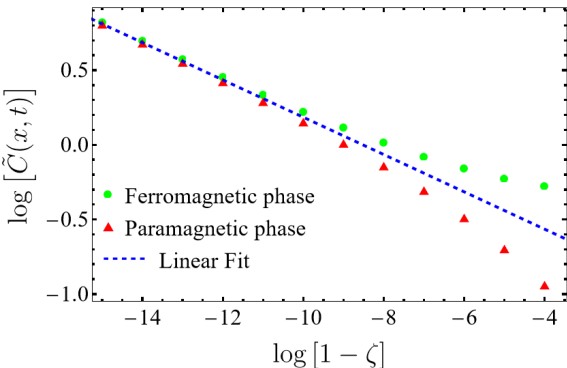

Figure 6: Near light cone behavior. Correlation funtions at $m\beta = 1$ and $mx = 1$ as a function of $1 - \zeta$. The green dashed line is a linear fit with slope $-1/8$.

We emphasize that recovering the conformal short-distance behavior would require the calculation of many terms in the form factor expansion, so the results in Fig. 4 demonstrate that the evaluation of the Fredholm determinant indeed amounts to a resummation of the form factor series. Moreover, the $\zeta > 1$ results provide an important crosscheck of the validity of our time-like algorithm and extrapolation scheme.

We close this section by looking at the behavior of the correlation function near the light cone. In the absence of Lorentz invariance, this is not equivalent to short separation. We present numerical results in Fig. 6 for $m\beta = 1$ at $mx = 1$ as a function of $1 - \zeta$, the distance from the light cone. We find evidence for a power law divergence with exponent $-1/8$ but with a temperature-dependent prefactor:

$$C_{\text{f/p}}(x, \zeta x; \beta) \xrightarrow{\zeta \to 1} m^{1/4} \chi(mx, m\beta)(1 - \zeta)^{-1/8}, \tag{44}$$

where $\chi(mx, m\beta)$ is a nonsingular proportionality factor. It would be interesting to analyze how the $\chi(mx, m\beta)$ function depends on its arguments, but we leave this to future work.

## 5.2 High temperatures

At temperatures much larger than the mass, $T \gg m$, the effects of the mass gap are negligible, and the correlators should follow the finite temperature conformal field theory predictions. Analytically continuing the conformal result [41] from imaginary to real time yields

$$C_{\text{f/p}}(x, t; \beta) \xrightarrow{m\beta \to 0} \left(\frac{\pi}{\beta}\right)^{1/4} \frac{2^{-1/6} e^{1/4} \mathcal{A}^{-3}}{\left[\sinh\left(\frac{\pi}{\beta}(x + t)\right) \sinh\left(\frac{\pi}{\beta}(x - t)\right)\right]^{1/8}}. \tag{45}$$

Note that for $x \pm t \ll \beta$, we recover the zero temperature result (37), because then the separation is again smaller than both $\beta$ and $1/m$.

In Fig. 7, we show the static correlation function at $m\beta = 0.1$ and $m\beta = 0.03$ inverse temperatures alongside with the prediction of Eq. (45). As can be seen, the correlators are close to the conformal result, and as the temperature increases, this approximation becomes more accurate.

To check the formula at finite $t$, we present the dynamical correlator in Fig. 8 for a fixed $mx = 0.005$ spatial separation at $m\beta = 0.005$ inverse temperature as a function of $\zeta$ in the space-like region. We can see that our numerical results are in agreement with the conformal expression (45). In particular, near the light cone $\zeta = 1$, the correlation functions diverge with a power law $\sim (1 - \zeta)^{-1/8}$. This is in agreement with the more general Eq. (44) where now the prefactor $\chi(mx, m\beta)$ follows from Eq. (45).

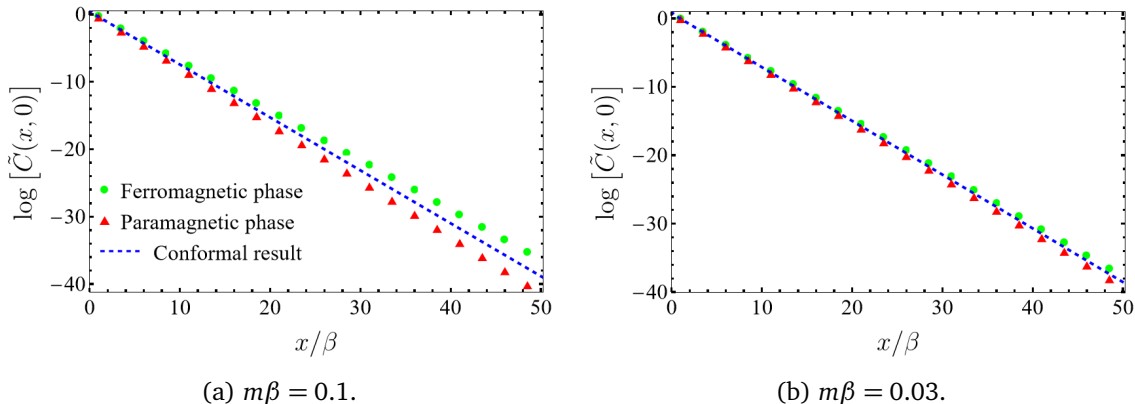

Figure 7: Static correlation functions at high temperatures. The blue dashed lines represent the conformal expression (45).

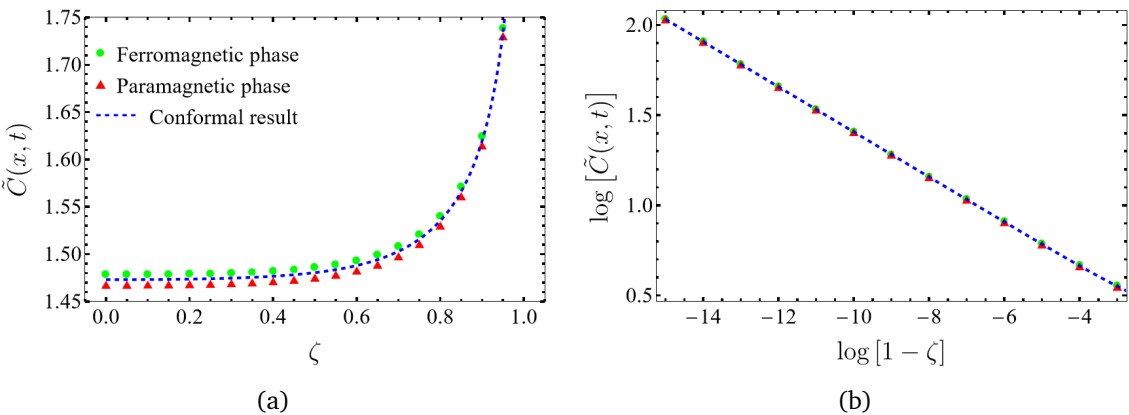

Figure 8: High temperature dynamical correlation functions for $\beta = 0.005$ and $x = 0.005$ as a function of $\zeta = t/x$. The blue dashed line represents Eq. (45).

Finally, let us discuss the high-temperature behavior of the time-like correlators. We plot the real part of the correlators for $\zeta = 1.5$ as functions of $mx$ in Fig. 9. It can be seen that as the temperature increases, the difference between the phases vanishes and the time-like correlators are also governed by Eq. (45).

### 5.3 Asymptotic behavior of the finite temperature correlations

Let us now turn to the asymptotic behavior of the correlation functions for large separations along space-time rays, i.e. in the limit $mx, mt \to \infty$ with $\zeta = t/x$ fixed. We follow the direction of increasing $\zeta$: we start with the $\zeta = 0$ equal-time case, then we proceed to the space-like $\zeta < 1$ domain, and finish our discussion with the $\zeta > 1$ time-like region.

#### 5.3.1 Decay of the equal-time correlations

We start our investigations with the static case, where there are exact analytical results for the asymptotics. The large distance behavior was derived in Ref. [42] using the Jordan–Wigner transformation and by evaluating the asymptotic behavior of the determinant of a Toeplitz matrix using the Szegő lemma. The result is a purely exponential decay,

$$C_{\text{f/p}}(x, 0; \beta) \xrightarrow{x \to \infty} \beta^{-1/4} g(\pm m\beta) e^{-x/\beta f(\pm m\beta)}, \tag{46}$$

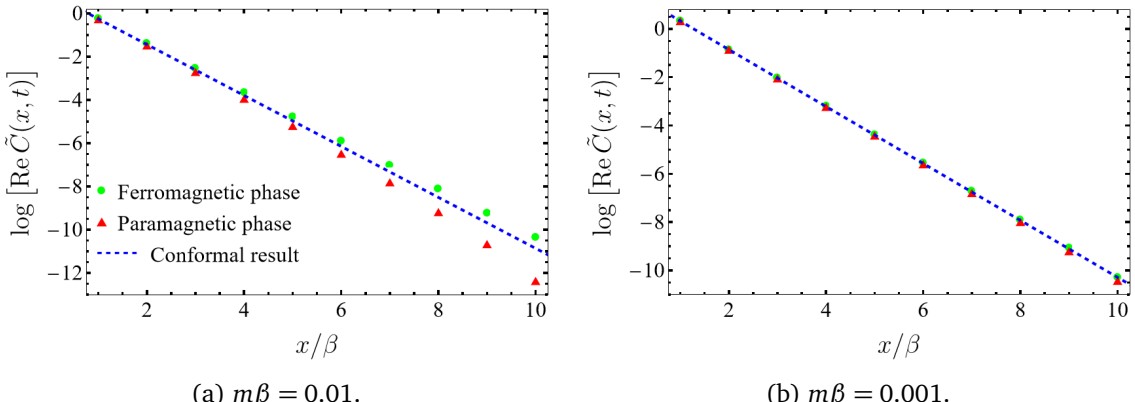

(a) $m\beta = 0.01$.                          (b) $m\beta = 0.001$.

Figure 9: High temperature simulation results for the time-like $\zeta = 1.5$ plotted on log-lin scale. The blue dashed line represents Eq. (45).

where the upper and lower signs correspond to the ferromagnetic and paramagnetic phases. The functions $f(s)$ and $g(s)$ are given by

$$f(s) = \int_0^\infty \frac{dy}{\pi} \log \coth\left(\frac{1}{2}\sqrt{s^2 + y^2}\right) + |s|\Theta(-s),\tag{47}$$

and

$$g(s) = \exp\left[\int_s^1 \frac{dy}{y}\left(f'(y)^2 - \frac{1}{4}\right) + \int_1^\infty \frac{dy}{y} f'(y)^2\right].\tag{48}$$

We note that for the ferromagnetic and paramagnetic phase $f(m\beta) = \beta\Delta\mathcal{E}(\beta)$ and $f(-m\beta) = \beta\Delta\mathcal{E}(\beta) + m\beta$, respectively (c.f. Eq. (24)). Moreover, we confirmed numerically the nontrivial integral identity $g(m\beta) = (m\beta)^{1/4}S(m\beta)^2$, where $S(m\beta)$ is defined in Eq. (23).

Note that the pure exponential decay in Eq. (46) seemingly contradicts Eqs. (41) and (43), but it is not the case since the $m\beta \to \infty$ and $mx \to \infty$ limits do not commute. This is trivial in the ferromagnetic phase, as the correlation function approaches a constant at zero temperature due to the presence of order, while it decays to zero at any finite temperature. This is however reflected by the fact that the correlation length diverges as $m\beta \to \infty$. But in the paramagnetic case, even though the correlation length approaches $1/m$, the zero temperature asymptotic behavior has an extra $\sim (mx)^{-1/2}$ power-law on top of the exponential decay which is missing at finite temperature.

In Fig. 10, we plot the static ($t = 0$) correlators for three different temperatures along with the prediction of Eq. (46). In all cases, our results match the asymptotic formula (46) perfectly even at relatively small $mx$ distances.

### 5.3.2 Asymptotic behavior in the space-like region

We proceed by discussing the asymptotic behavior of the dynamical correlation functions at large space-like separations.

The various analytical approaches all predict that in the ferromagnetic region, the time dependence of the asymptotics comes from a factor [8, 9, 22, 24]

$$\sim \exp\left(-\int dp\, f(p;\beta)|x - v(p)t|\right),\tag{49}$$

where $v(p) = p/\varepsilon(p)$ is the velocity. Since $|v(p)| < 1$ is bounded by the speed of light, $|x - v(p)t| = |x|$ for spatial separations, $|x| > |t|$, implying that the time-dependence disappears. Our numerical analysis confirmed that the asymptotic behavior in the ferromagnetic

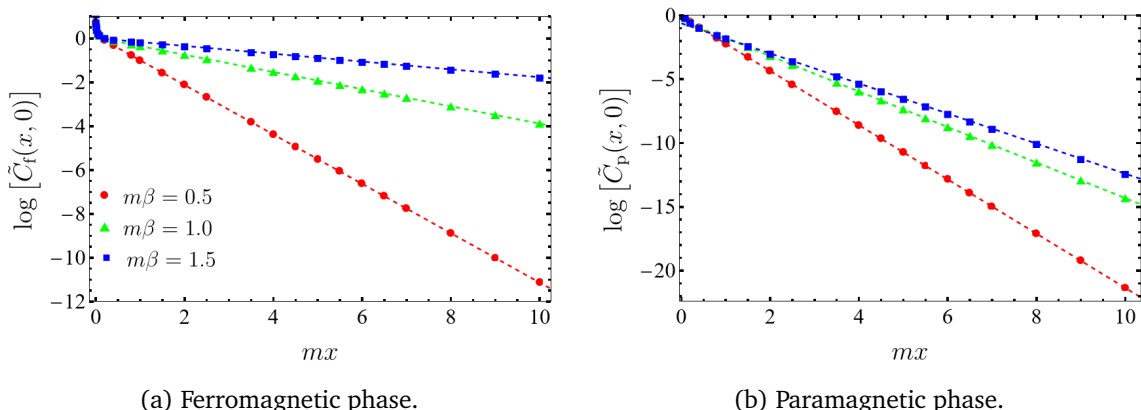

(a) Ferromagnetic phase.

(b) Paramagnetic phase.

Figure 10: Asymptotic behavior of static correlation functions at three different temperatures. The dashed lines represent the analytical result (46).

phase is independent of $0 \leq \zeta < 1$, so it is given by Eq. (46). Therefore, we focus on the paramagnetic case where, as we shall see, the asymptotic behavior has a nontrivial $\zeta$-dependence.

Since for space-like separations, Eq. (22) is a large-distance and large-time expansion, the asymptotic behavior can be obtained by its first term. This implies that for large separations, the paramagnetic correlation functions can be calculated as

$$C_{\mathrm{p}}(x,t;\beta) \xrightarrow{x,t\to\infty} C_{\mathrm{p}}^{\mathrm{asym}}(x,t;\beta) = m^{1/4}S(m\beta)^2 e^{-\Delta\mathcal{E}x}$$
$$\times \int_{-\infty}^{\infty} \frac{d\theta}{2\pi} \left( \frac{e^{im(x\sinh\theta - t\cosh\theta)}}{1 - e^{-m\beta\cosh\theta}} e^{\eta(\theta)} + \frac{e^{-im(x\sinh\theta - t\cosh\theta)}}{e^{m\beta\cosh\theta} - 1} e^{-\eta(\theta)} \right). \quad (50)$$

Our task is now to extract the asymptotic behavior of this integral. It turns out that it can be done by tracing back our steps we took to derive the form factor series in Appendix A. Changing variables in the second term as $\theta = \vartheta + i\pi$ allows us to write the two separate integrals as a single complex contour integral:

$$C_{\mathrm{p}}^{\mathrm{asym}}(x,t;\beta) = m^{1/4}S(m\beta)^2 e^{-\Delta\mathcal{E}x} \oint_C \frac{d\theta}{2\pi} \frac{e^{im(x\sinh\theta - t\cosh\theta) + \eta(\theta)}}{1 - e^{-m\beta\cosh\theta}}, \quad (51)$$

where the contour $C$ is a counterclockwise curve consisting of the $\mathrm{Im}(\theta) = \delta$ and $\mathrm{Im}(\theta) = \pi - \delta$ lines. The poles of the integrand lie along the $\mathrm{Im}\theta = \pi/2$ line at

$$\theta_k = \sinh^{-1}\left(\frac{2\pi}{m\beta}k\right) + i\pi/2, \qquad k \in \mathbb{Z}. \quad (52)$$

Using the residue theorem,

$$\oint_C \frac{d\theta}{2\pi} \frac{e^{im(x\sinh\theta - t\cosh\theta) + \eta(\theta)}}{1 - e^{-m\beta\cosh\theta}} = 2\pi i \sum_{k=-\infty}^{\infty} \frac{1}{2\pi} \frac{e^{im(x\sinh\theta_k - t\cosh\theta_k) + \eta(\theta_k)}}{m\beta\sinh(\theta_k)e^{-m\beta\cosh\theta_k}}, \quad (53)$$

we obtain the following, more convenient representation:

$$C_{\mathrm{p}}^{\mathrm{asym}}(x,t;\beta) = m^{1/4}S(m\beta)^2 e^{-\Delta\mathcal{E}x} \sum_{k=-\infty}^{\infty} \frac{\exp\left[-x\sqrt{m^2 + q_k^2} + tq_k\right]e^{2\kappa(q_k)}}{\beta\sqrt{m^2 + q_k^2}}, \quad (54)$$

where $q_k = 2\pi k/(m\beta)$ and we used that $\eta\left(\sinh^{-1}(q) + i\pi/2\right) = 2\kappa(q)$ where $\kappa(q)$ is defined in Eq. (A.5). This is a sum of decaying exponentials (recall that $|x| > |t|$), so the asymptotic behavior for a given $\zeta$ and $m\beta$ will be given by the slowest decaying exponential,

$$C_{\mathrm{p}}^{\mathrm{asym}}(x, t; \beta) = m^{1/4} S(m\beta)^2 e^{-\Delta\mathcal{E}x} \frac{\exp\left[-x\sqrt{m^2 + q_{k_0}^2} + t q_{k_0}\right] e^{2\kappa(q_{k_0})}}{\beta\sqrt{m^2 + q_{k_0}^2}}, \tag{55}$$

where

$$k_0 = \operatorname*{argmin}_{k \in \mathbb{Z}}\left[\sqrt{1 + \frac{4\pi^2}{m^2\beta^2}k^2} - \zeta\frac{2\pi}{m\beta}k\right]. \tag{56}$$

Importantly, this means that since $k_0$ can only take discrete (integer) values, the correlation length given by

$$\xi(\zeta; \beta) = \left[\sqrt{m^2 + \frac{4\pi^2}{\beta^2}k_0^2 - \zeta\frac{2\pi}{\beta}k_0} + \Delta\mathcal{E}(\beta)\right]^{-1}, \tag{57}$$

is *not an analytic function* of $\zeta$ and $\beta$: it is continuous, but at specific parameters it has cusps. To demonstrate this, in Fig. 11 we plot $\xi(\zeta; \beta)$ for $m\beta = 5$ as a function of $\zeta$, and for $\zeta = 0.5$ as a function of $m\beta$.

The location of the cusps can be determined from the condition that the argument in Eq. (56) is equal at two consecutive integers, $k = \ell - 1$ and $k = \ell$. For $\beta$ fixed, this gives

$$\zeta_\ell = \sqrt{\ell^2 + \frac{m^2\beta^2}{4\pi^2}} - \sqrt{(\ell - 1)^2 + \frac{m^2\beta^2}{4\pi^2}}, \qquad \ell = 1, 2, \ldots, \tag{58}$$

for the rays where the cusps appear. For $\ell \ll m\beta/(2\pi)$, they are evenly spaced, $\zeta_\ell \approx (2\ell - 1)\pi/\beta$, while as $\ell \to \infty$, $\zeta_\ell \approx 1 - m^2\beta^2/(8\pi^2\ell^2)$, showing that there are infinitely many cusps accumulating as $\zeta \to 1$. For a fixed ray $\zeta$, the cusps are at

$$m\beta_\ell = \frac{\pi}{\zeta}\sqrt{(2\ell - 1)^2 - \zeta^2}\sqrt{1 - \zeta^2}, \qquad \ell = 1, 2, \ldots \tag{59}$$

As $\ell \to \infty$, they become evenly spaced, $m\beta_\ell \approx (2\ell - 1)\sqrt{1 - \zeta^2}\,\pi/\zeta$.

Interestingly, besides the cusps, the correlation length is not monotonically decreasing as a function of temperature (see Fig. 11b). This counterintuitive behavior takes place for

$$2\pi\ell\sqrt{1 - \zeta^2}/\zeta < m\beta < \pi\sqrt{(2\ell + 1)^2 - \zeta^2}\sqrt{1 - \zeta^2}/\zeta, \qquad \ell = 1, 2, \ldots \tag{60}$$

At $\zeta = 0$, the minimum is at $k_0 = 0$, and the inverse correlation length is simply $\xi(0; \beta)^{-1} = \Delta\mathcal{E}(\beta) + m$ in agreement with Eq. (47). Comparing the full expressions in Eqs. (46) and (55) and recalling that $S(m\beta)^2 = (m\beta)^{-1/4}g(m\beta)$, we obtain the nontrivial relation

$$g(m\beta)/g(-m\beta) = m\beta e^{-2\kappa(0)} = m\beta e^{-\eta(i\pi/2)}, \tag{61}$$

where $g(s)$, $\kappa(q)$, and $\eta(\theta)$ are defined in Eqs. (48), (A.5), and (25), respectively.

Upon increasing $\zeta$ at a fixed $\beta$, $k_0 = 0$ remains the solution of Eq. (56) up to some $\zeta_1$ (see Fig. 11a), which means that the asymptotic behavior of the dynamical correlation function will be the same as the static one for $0 \le \zeta \le \zeta_1$. When approaching the light cone, $k_0 \to \infty$, the cusps become suppressed. Then the discreteness of $k_0$ can be neglected in the minimization (56), and the inverse correlation length becomes the relativistic $m\sqrt{1 - \zeta^2}$. The same happens as $m\beta \to \infty$ for a fixed $\zeta$ (see the blue dashed lines in Fig. 11).

To check the validity of Eq. (55), we numerically calculated the paramagnetic correlation functions at temperatures $m\beta = 5$ and $m\beta = 1$ for four different rays; these results are depicted in Fig. 12. We can see that the simulated data matches the theoretical results perfectly.

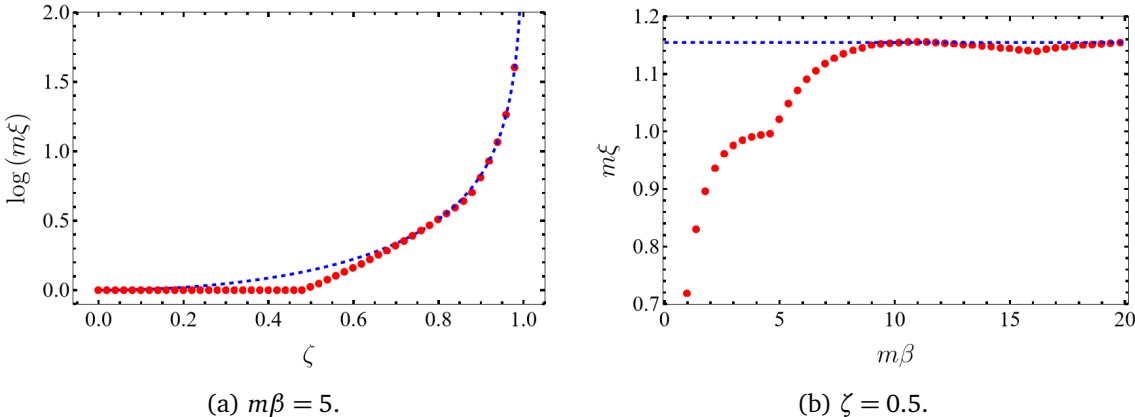

(a) $m\beta = 5$.         (b) $\zeta = 0.5$.

Figure 11: Theoretical result for the non-analytic correlation length of the paramagnetic correlation functions in the space-like region (a) at fixed $m\beta = 5$ as a function of $\zeta = t/x$; (b) at a fixed ray $\zeta = 0.5$ as a function of $m\beta$. The blue dashed line shows the function $(1 - \zeta^2)^{-1/2}$.

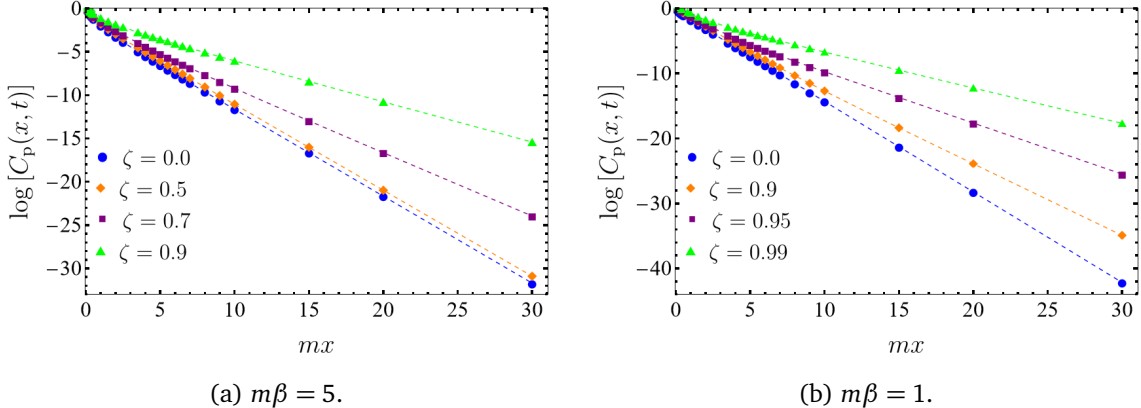

(a) $m\beta = 5$.         (b) $m\beta = 1$.

Figure 12: Numerical results (dots) for the dynamical correlation functions in the paramagnetic phase at inverse temperatures $m\beta = 5$ and $m\beta = 1$. Different colors indicate different $\zeta = t/x$ rays. The dashed lines represent the asymptotic Eq. (55).

Note that at $m\beta = 5$, the asymptotic correlators differ from the static ($\zeta = 0$) curve only above $\zeta \approx 0.5$, in harmony with the first cusp in Fig. 11a, while for $m\beta = 1$, this threshold increases to $\zeta \approx 0.85$. This is a general feature of Eq. (57): the effects of a temporal separation are only significant at low temperatures; at high temperatures, only those regions are affected that are close to the light cone.

This finding is different from the available theoretical results in the literature [8, 9, 24]. These are not exactly the same but they have the same semiclassical low-temperature limit, and share the general form

$$C_{\mathrm{p}}(x, \zeta; \beta) \xrightarrow{x \to \infty} A(x, \zeta; \beta) e^{-x/\ell(\beta)}. \tag{62}$$

Here $\ell(\beta)$ is either equal to $\Delta\mathcal{E}(\beta)$ or has the same low-temperature behavior, and $A(x, \zeta; \beta)$ is the zero-temperature result with or without an additive finite temperature correction. This expression predicts a correlation length that depends continuously on both $\beta$ and $\zeta$. Due to the exponentially decaying $A(x, \zeta; \beta)$, the correlation length differs from the $t = 0$ one for any $\zeta > 0$, and there is a $\sim x^{-1/2}$ decay besides the exponential.

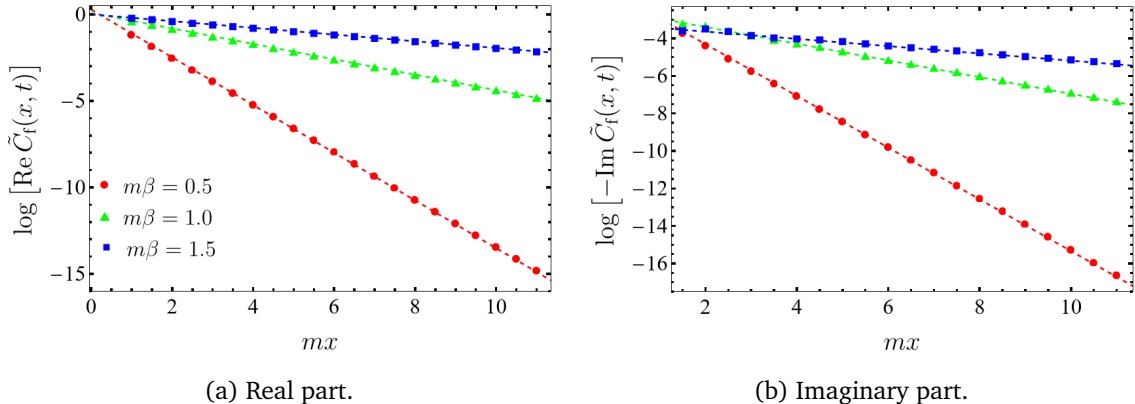

(a) Real part.  (b) Imaginary part.

Figure 13: Asymptotic behavior of the dynamical correlation functions at the time-like $\zeta = 1.5$ space-time ray in the ferromagnetic phase plotted on log-lin scale for three different temperatures. The dashed lines represent the best fits with fixed slope given by the correlation length (65).

To understand the discrepancy better, consider the low temperature expansion of the integrand of Eq. (50), leading to[3]

$$m^{1/4}e^{-\Delta\mathcal{E}x}\frac{1}{\pi}\left[K_0\left(m\sqrt{x^2-t^2}\right)+\int_{-\infty}^{\infty}d\theta\,\cos(mt\cosh\theta-mx\sinh\theta)e^{-m\beta\cosh\theta}\right].\quad(63)$$

This is of the form (62) and it agrees with the scaling limit of the lattice result of Ref. [8]. As discussed above, its asymptotic behavior is not given by Eq. (55), which shows that the $m\beta \to \infty$ and $x \to \infty$ limits do not commute. In this sense, our results correspond to the large $x$ behavior for a fixed $\beta$ and $\zeta$, while that of Ref. [8, 9, 24] corresponds to the small temperature behavior for a fixed large $x$ and $t$.

### 5.3.3 Asymptotic behavior in the time-like region

Finally, we analyze the asymptotic characteristics of the two-point function for time-like separations. Even though the original series (22) is not well-defined in this domain, our numerical method discussed in Sec. 3.2.1 allows us to extract physically meaningful results. We compare our numerics to theoretical predictions available in the literature. One of our main references is the work [8] which performed an elaborate form factor calculation, combined with the representative state method, in the quantum Ising spin chain.

First, we discuss the ferromagnetic phase. Let us start by quoting the available theoretical results in the literature. Taking the scaling limit of the corresponding result of Ref. [8] we obtain

$$C_{\mathrm{f}}(x,\zeta;\beta)\xrightarrow{x\to\infty}\tilde{C}(\beta)\exp\left(-x/\xi(\zeta,\beta)\right),\quad(64)$$

where $\tilde{C}(\beta)$ is an undetermined temperature dependent constant and

$$\xi(\zeta,\beta)^{-1}=\int_{-\infty}^{\infty}\frac{dp}{2\pi}\log\coth\left[\frac{\beta\sqrt{m^2+p^2}}{2}\right]\left|1-\frac{\zeta p}{\sqrt{m^2+p^2}}\right|,\quad(65)$$

is the inverse correlation length. In the low-temperature limit, this agrees with the semiclassical result of Ref. [9] and with the result of Ref. [22].

---

[3]Here we used that as $m\beta \to \infty$, $\eta(\theta) \to 0$ and $S(m\beta) \to 1$.

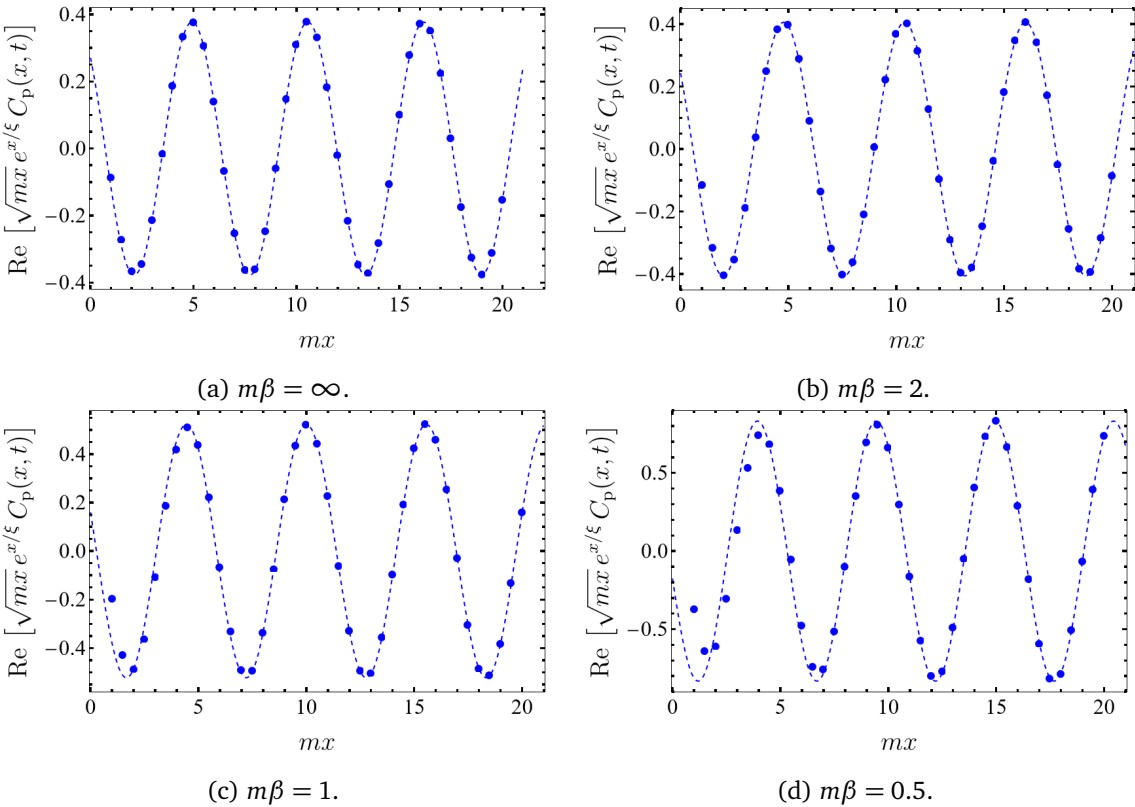

(a) $m\beta = \infty$.

(b) $m\beta = 2$.

(c) $m\beta = 1$.

(d) $m\beta = 0.5$.

Figure 14: Asymptotic behavior of the time-like paramagnetic correlation functions for $\zeta = 1.5$. The blue dots represent the simulated correlators divided by the exponential decay of Eq. (64) and an additional square root factor. The dashed lines are harmonic fits of the form $A\sin(\omega x + \phi)$.

We computed the ferromagnetic correlators for $\zeta = 1.5$ for three different temperatures with the results shown in Fig. 13. Here the dashed lines represent the best linear fits with slopes given by Eq. (65). As it is clear from the plots, the dataset matches these fits perfectly well. The offsets of the fitted lines would allow us to numerically determine the unknown amplitude $\tilde{C}(\beta)$.

We now turn to the asymptotics of the paramagnetic correlator for time-like separations. Here the results of Ref. [8] state that the correlation length is the same as in the ferromagnetic phase, but the amplitude has a nontrivial $x$-dependence. It is a natural assumption [8, 9, 24] that at low temperature, it is close to the zero-temperature result Eq. (39), but its temperature dependence is currently unknown.

To test these claims and to investigate this unknown feature, we performed numerical simulations at $\zeta = 1.5$ and at four different temperatures. In Fig. 14 we show the real part[4] of the correlators divided by the exponential decay in Eq. (64) and by an additional $\sqrt{mx}$ factor motivated by the asymptotic behavior (41a) of the zero temperature Bessel function. As can be seen in the plots, the datasets shown in blue dots describe oscillations of constant amplitude, providing evidence for a subleading square root decay on top of the exponential decay in Eq. (64). The frequency $\omega$ extracted from the data is close to the zero temperature value $m\sqrt{\zeta^2 - 1}$ coming from the Bessel function. We observed a slight temperature dependence of the frequency, but at present, we cannot give any definite claims about this. We leave the investigation of this question to the future.

---

[4]The imaginary parts can be treated identically and give very similar results.

# 6 Summary and outlook

In this work, we studied the zero and finite temperature dynamical correlation functions of the magnetization operator in the one-dimensional Ising quantum field theory. Our approach was based on the finite temperature form factor series of Refs. [7,22] which, due to the special structure of the form factors, allowed for a Fredholm determinant representation of the correlators amenable to numerical evaluation. Even though the form factor series in its original form is well-defined only for space-like separations, we developed a method that allowed us to compute correlation functions at time-like separations, at least for not too large $\zeta = t/x$ rays. This method was based on the analytic continuation of the $\zeta$ parameter to complex values and involved an extrapolation to the physical point.

Using this numerical technique, we were able to explore, remarkably, all space-time and temperature regimes in both phases of the theory. We benchmarked our method with zero-temperature calculations as well as by recovering the conformal field theory behavior at small separations and high temperatures. We found that the analytic continuation from imaginary to real times works even in the time-like regime. Then we investigated the behavior of correlations near the light cone and at large separations, where several analytical predictions were available. Apart from the space-like paramagnetic correlator, we found perfect agreement with the scaling limit of the asymptotic results of Ref. [8] derived in the spin chain. Since in the low-temperature limit, these expressions agree with other approaches [9,22,24], we confirmed these predictions as well.

The only exception where we found an unexpected deviation from these predictions was the paramagnetic space-like correlation function. Here we analytically obtained a purely exponential decay along each ray defined by $\zeta = t/x$. Surprisingly, the correlation length has some unusual characteristics. Namely, it is a non-analytic function of both the direction $\zeta$ and the temperature $m\beta$ (c.f. Fig. 11). Similar non-analytic behaviors were observed for equal time correlators in the XXZ spin chain [43], the t-J model [44], and the Lieb-Liniger model [45] using the quantum transfer matrix approach. Moreover, the temperature dependence of the correlation length is non-monotonic, and rather counter-intuitively, it can increase with increasing temperature. We also pointed out that limits of large separation and low temperature do not commute in this case. To the best of our knowledge, these features have not been noticed before in the literature.

In the future, we would like to investigate this particular case in the spin chain. Our preliminary numerical results show that the unusual features are not peculiarities of the field theory but are also shared by the spin chain. It may be possible to derive the asymptotic behavior using the ballistic fluctuation approach of Refs. [46–48]. It would also be interesting to extend the numerical approach to the entire time-like domain and to investigate whether the analytic continuation of $\zeta$, applied in our numerical algorithm, can also be used to derive analytical results.

## Acknowledgments

We are grateful to Benjamin Doyon and Gábor Takács for insightful discussions.

**Funding information** This work was supported by the National Research, Development and Innovation Office of Hungary (NKFIH) through the OTKA Grant K138606. I. Cs. was also partially supported by the ÚNKP-23-2-I-BME-120 New National Excellence Program of the Ministry for Culture and Innovation from the source of the National Research, Development and Innovation Fund.

# A Derivation of the finite temperature form factor series

In this Appendix, we provide the derivation of the infinite series representation of the correlation function. Our discussion is based on Refs. [7, 22].

The idea is to focus on the correlator in imaginary time. The geometry of the Euclidean space-time is that of a cylinder, infinite in the spatial ($x$) direction and periodic in the imaginary time ($\tau$) direction with period $\beta$, the inverse temperature. Exploiting relativistic invariance, the correlation function can be calculated by swapping the roles of space and time, when the cylinder geometry corresponds to the same field theory in finite volume $\beta$ with periodic boundary conditions and, importantly, at zero temperature. The correlation function can be computed via a zero-temperature form factor expansion using the exact finite volume form factors that can be obtained from the corresponding lattice results [35–37] but were also computed in the field theory in Ref. [38]. Here we only need the elementary form factors where one of the states is the ground state. We recall from Sec. 2 that in finite volume, the ground state is always in the Neveu–Schwarz (NS) sector and $\hat{\sigma}$ has nonzero matrix elements between states belonging to different sectors. If the volume is $\beta$,

$$
{}_\mathrm{R}\langle k_1, \ldots, k_N | \hat{\sigma} | 0 \rangle_\mathrm{NS} = i^{[N/2]} m^{1/8} S(m\beta) \prod_{j=1}^N \frac{e^{\kappa(q_{k_j})}}{\sqrt{\beta \varepsilon(q_{k_j})}} \prod_{i<j} \frac{q_{k_i} - q_{k_j}}{\varepsilon(q_{k_i}) - \varepsilon(q_{k_j})}, \tag{A.1}
$$

where $q_k = 2\pi/\beta \, k$ with $k$ integer (Ramond sector),

$$
S(m\beta) = \exp\left[ \frac{\beta^2}{2\pi^2} \int_0^\infty \frac{dp_1 \, \omega'(p_1)}{\sinh(\beta \omega(p_1))} \int_0^\infty \frac{dp_2 \, \omega'(p_2)}{\sinh(\beta \omega(p_2))} \log\left| \frac{p_1 + p_2}{p_1 - p_2} \right| \right], \tag{A.2}
$$

and

$$
\kappa(q) = \frac{\varepsilon(q)}{\pi} \int_0^\infty \frac{dp}{\varepsilon(q)^2 + p^2} \log \tanh\left( \frac{\beta \varepsilon(p)}{2} \right). \tag{A.3}
$$

In terms of the rapidity variables, $q_k = m \sinh\theta_k$, $\varepsilon(q_k) = m \cosh(\theta_k)$, and

$$
{}_\mathrm{R}\langle k_1, \ldots, k_N | \hat{\sigma} | 0 \rangle_\mathrm{NS} = i^{[N/2]} m^{1/8} S(m\beta) \prod_{j=1}^N \frac{e^{-\tilde{\eta}(\theta_{k_j})/2}}{\sqrt{m\beta \cosh(\theta_{k_j})}} \prod_{i<j} \tanh\left( \frac{\theta_{k_i} - \theta_{k_j}}{2} \right), \tag{A.4}
$$

where

$$
\tilde{\eta}(\theta) = \int_{-\infty}^\infty \frac{d\theta'}{\pi} \frac{1}{\cosh(\theta - \theta')} \log\left[ \frac{1 + e^{-m\beta \cosh\theta'}}{1 - e^{-m\beta \cosh\theta'}} \right], \tag{A.5}
$$

and $S(m\beta)$ is alternatively given by Eq. (23).

A two-point function on the cylinder can be computed with the zero-temperature form factor expansion

$$
\langle \hat{\sigma}(\tau, x) \hat{\sigma}(0, 0) \rangle = m^{1/4} S(m\beta)^2 e^{-\Delta\mathcal{E}(\beta)|x|} \sum_{N=0}^\infty \frac{1}{N!} \prod_{j=1}^N \sum_{k_j = -\infty}^\infty \frac{e^{-|x|\varepsilon_j - i\tau q_j + 2\kappa(q_j)}}{\beta \varepsilon_j} \prod_{i<j} \left( \frac{q_i - q_j}{\varepsilon_i + \varepsilon_j} \right)^2, \tag{A.6}
$$

where $q_j = q_{k_j}$ and $\varepsilon_j = \varepsilon(q_{k_j})$ and $\Delta\mathcal{E}(\beta)$, given in Eq. (24), is the energy difference of the Ramond and Neveu–Schwarz ground states. Notice the unusual products in the exponent which are due to the space and time being exchanged. The infinite sums over the $k_j$ integers can be converted into contour integrals. Consider the contour integral on the complex $\vartheta$-plane

$$
\oint_C \frac{d\vartheta}{2\pi} \frac{e^{-m\tau \cosh\vartheta + im|x| \sinh\vartheta - \tilde{\eta}(\vartheta - i\pi/2)}}{1 - e^{-m\beta \cosh\vartheta}}, \tag{A.7}
$$

where the contour $C$ consists of two parallel lines running below and above the $\text{Im}\vartheta = \pi/2$ line. The simple poles of the integrand come from the zeros of the denominator at $\vartheta_k = i\pi/2 + \theta_k$ where

$$m \sinh\theta_k = -i\, m \cosh\vartheta_k = \frac{2\pi}{\beta}k = q_k\,, \tag{A.8a}$$

$$m \cosh\theta_k = -i\, m \sinh\vartheta_k = \varepsilon(q_k)\,. \tag{A.8b}$$

Expanding the denominator around each pole to the first order, by the residue theorem we obtain

$$\oint_C \frac{d\vartheta}{2\pi} \frac{e^{-m\tau\cosh\vartheta + im|x|\sinh\vartheta - \tilde{\eta}(\vartheta - i\pi/2)}}{1 - e^{-m\beta\cosh\vartheta}} = \sum_{k=-\infty}^{\infty} \frac{e^{-|x|\varepsilon(q_k) - i\tau q_k + 2\kappa(q_k)}}{\beta\,\varepsilon(q_k)}\,. \tag{A.9}$$

Replacing the sum over each $k_j$ by such a contour integral leads to a product of integrals over the rapidity variables $\vartheta_1, \ldots, \vartheta_N$ (note that the index here is different from the $k$ index in Eqs. (A.8)). The double product will be recovered by the residue theorem from the factor

$$\prod_{i<j} \left( \frac{-i\, m \cosh\vartheta_i + i\, m \cosh\vartheta_j}{-i\, m \sinh\vartheta_i - i\, m \sinh\vartheta_j} \right)^2 = \prod_{i<j} \tanh\left( \frac{\vartheta_i - \vartheta_j}{2} \right)^2\,. \tag{A.10}$$

Pushing the upper contours to $\text{Im}\theta_j = \pi - \delta$ parameterized as $\vartheta_j = \theta_j + i\pi - i\delta$ turns the hyperbolic functions to $\sinh/\cosh(\vartheta_j) \to -\sinh/\cosh(\theta_j - i\delta)$ and the integral over the real $\theta_j$ receives a sign due to the direction of the contour. Similarly, pushing the lower contours to $\text{Im}\theta_j = \delta$ parameterized as $\vartheta_j = \theta_j + i\delta$ results in $\sinh/\cosh(\vartheta_j) \to \sinh/\cosh(\theta_j + i\delta)$. The contour shifts yield a pair of integrals over each $\theta_j$. After analytically continue to real time by setting $\tau = it$, we obtain the result (22), where $\eta(\theta) = \pm\tilde{\eta}(\theta \pm i\pi/2)$, and the sign $\epsilon = \pm$ indexing holes and particles corresponds to the choice of upper and lower contour.

# B  Correlation functions as Fredholm determinants

In this appendix, we show that the form factor expansions in Eqs. (18) and (22) can be represented as Fredholm determinants. First, we review the definition of Fredholm determinants and then show that their formalism matches that of the form factor expansions perfectly. It leads to new expressions for the correlation functions that give the foundation for an efficient numerical algorithm discussed in Sec. 3.2.

## B.1  Fredholm determinants

To develop the theory of Fredholm determinants, let us first consider the integral equation

$$u(\theta) + \int_a^b d\theta'\, K(\theta, \theta')u(\theta') = f(\theta)\,, \tag{B.1}$$

where $u(\theta)$ and $f(\theta)$ are complex functions and $K(\theta, \theta')$ is a complex kernel. This is the continuum version of a matrix equation for $u$, a viewpoint that soon becomes very useful for our purposes. In an abstract notation, we can write this as

$$(\mathbb{I} + \mathbf{K})u = f\,, \tag{B.2}$$

where $\mathbb{I}$ is the identity and $\mathbf{K}$ is an appropriate integral operator. Now, we can ask whether such an equation has a solution, i.e. what is the determinant of the $\mathbb{I} + \mathbf{K}$ transformation.

However, since it is not a finite-dimensional object, this calculation raises a few obstacles. To surpass these, the strategy is the following: we discretize Eq. (B.1) by splitting the $[a, b]$ interval into $N$ equal pieces, calculate the determinant in this finite space and then take the $N \to \infty$ limit. After the discretization, we get the following matrix equation:

$$\left( \sum_{j=1}^{N} \delta_{ij} + h \sum_{j=1}^{N} K(\theta_i, \theta_j) \right) u(\theta_j) = f(\theta_i), \tag{B.3}$$

where $h = (b-a)/N$. In this case, we can approximate the desired determinant with

$$D(N) = \det \left[ \delta_{ij} + h K(\theta_i, \theta_j) \right]. \tag{B.4}$$

As described above, we expect that $D(N) \to \det(\mathbb{I} + \mathbf{K})$ as $N \to \infty$.

Next, we can proceed by expressing $D(N)$ as a polynomial in $h$:

$$D(N) = \sum_{m=0}^{N} a_m h^m. \tag{B.5}$$

It turns out that the $a_m$ coefficients can be expressed as subdeterminants of the matrix $K_{ij} \equiv K(\theta_i, \theta_j)$ with the following identity:

$$D(N) = 1 + h \sum_{i=1}^{N} |K_{ii}| + \frac{h^2}{2} \sum_{i,j=1}^{N} \begin{vmatrix} K_{ii} & K_{ij} \\ K_{ji} & K_{jj} \end{vmatrix} + \frac{h^3}{6} \sum_{i,j,k=1}^{N} \begin{vmatrix} K_{ii} & K_{ij} & K_{ik} \\ K_{ji} & K_{jj} & K_{jk} \\ K_{ki} & K_{kj} & K_{kk} \end{vmatrix} + \dots \tag{B.6}$$

Now, we can easily take the $N \to \infty, h \to 0$ limit of this expansion to arrive at

$$\det(\mathbb{I} + \mathbf{K}) = \sum_{k=0}^{\infty} \int_a^b \frac{d\theta_1 \dots d\theta_k}{k!} \det \left[ K(\theta_i, \theta_j) \right]_{i,j=1}^{k}. \tag{B.7}$$

This can be viewed as the definition of a Fredholm determinant. Notice the similarity between this result and the zero temperature form factor expression (18). We would arrive at a Fredholm determinant representation of the correlators if we could find such kernel functions that generate the form factors as their subdeterminants. In the next section, we will do precisely that, but first, we have to generalize the theory a bit further until it becomes applicable to finite temperatures.

The Fredholm determinant representations differ between zero and finite temperatures due to an extra summation for the $\epsilon$ signs in Eq. (22). However, we can adapt our previous formalism to this case, too, by slightly modifying the initial integral equation:

$$u_\epsilon(\theta) + \sum_{\epsilon'=\pm} \int_a^b d\theta' K_{\epsilon,\epsilon'}(\theta, \theta') u_{\epsilon'}(\theta') = f_\epsilon(\theta). \tag{B.8}$$

It is reminiscent of a matrix equation of size $2N$, so it is natural to introduce the notations $\tilde{u} = (\{u_+(\theta)\}, \{u_-(\theta)\})$, $\tilde{f} = (\{f_+(\theta)\}, \{f_-(\theta)\})$, $\xi = (\{\theta_i\}, \{\theta_i\})$ and

$$\tilde{K} = \begin{pmatrix} \{K_{++}(\theta_i, \theta_j)\} & \{K_{+-}(\theta_i, \theta_j)\} \\ \{K_{-+}(\theta_i, \theta_j)\} & \{K_{--}(\theta_i, \theta_j)\} \end{pmatrix}. \tag{B.9}$$

In this language, the discretized equation becomes

$$\tilde{u}(\xi_i) + h \sum_{j=1}^{2N} \tilde{K}(\xi_i, \xi_j) \tilde{u}(\xi_j) = \tilde{f}(\xi_i), \tag{B.10}$$

which means that we can apply Eq. (B.6) to get the following approximation for the Fredholm determinant:

$$D(N) = 1 + h \sum_{i=1}^{2N} |\tilde{K}_{ii}| + \frac{h^2}{2} \sum_{i,j=1}^{2N} \begin{vmatrix} \tilde{K}_{ii} & \tilde{K}_{ij} \\ \tilde{K}_{ji} & \tilde{K}_{jj} \end{vmatrix} + \dots \tag{B.11}$$

Now, using the fact that for a general $g$ function

$$\sum_{i=1}^{2N} \tilde{g}(\xi_i) = \sum_{\epsilon=\pm} \sum_{i=1}^{N} g_\epsilon(\theta_i), \tag{B.12}$$

we get the final result

$$\det(\mathbb{I} + \tilde{\mathbf{K}}) = \sum_{k=0}^{\infty} \sum_{\epsilon_1, \dots \epsilon_k = \pm} \int_a^b \frac{d\theta_1 \dots d\theta_k}{k!} \det\left[ K_{\epsilon_i, \epsilon_j}(\theta_i, \theta_j) \right]_{i,j=1}^k, \tag{B.13}$$

which has exactly the desired structure. As we will see in the next section, this result gives the foundations for the Fredholm determinant representation of the finite temperature correlation functions.

## B.2  Representing the correlation functions as Fredholm determinants

As previously noted, there is a striking similarity between the formalism of Fredholm determinants and form factor expansions. When we align the kernel functions with the form factors, we can discover Fredholm determinant representations for the correlation functions, which proves highly beneficial for numerical analysis. In this section, we carry out the computations necessary for this purpose.

First, let us consider the zero temperature case, as it has a much simpler formalism. We would like to give Fredholm determinant representations for Eqs. (18) using Eq. (B.7). Following [7, 26], it is useful to introduce the

$$C_\pm(x,t) = C_{\mathrm{f}}(x,t) \pm C_{\mathrm{p}}(x,t)$$
$$= m^{1/4} \sum_{k=0}^{\infty} \frac{(\pm 1)^k}{k!} \prod_{i=1}^{k} \left( \int_{-\infty}^{\infty} \frac{d\theta_i}{2\pi} e^{imx \sinh\theta_i - imt \cosh\theta_i} \right) \prod_{1 \le i < j \le k} \tanh^2\left( \frac{\theta_i - \theta_j}{2} \right) \tag{B.14}$$

combinations, since it removes any restrictions from the domain of $k$. Here, however, we have to deal with a little complication. While this formula is perfectly well-defined, during a numerical analysis, it fails to converge since the exponential terms oscillate very fast for large rapidities. However, as discussed in Sec. 3.1, by extending the rapidities to the complex plane in a clever way, we can circumvent these issues, although we have to use different contours for space-like and time-like separated $(x, t)$ points. For space-like separations, a simple contour shift will suffice,

$$\Gamma : \mathbb{R} \to \mathbb{C}, \quad \theta \to \theta + i\delta, \tag{B.15}$$

while for time-like separations we need to use an appropriate rapidity-dependent shift, e.g.

$$\Gamma : \mathbb{R} \to \mathbb{C}, \quad \theta \to \theta - i\delta \tanh(\theta). \tag{B.16}$$

In both cases, $\delta$ is a sufficiently small positive constant. Implementing these contour shifts, we get the following result:

$$C_\pm(x,t) = m^{1/4} \sum_{k=0}^{\infty} \frac{(\pm 1)^k}{k!} \prod_{i=1}^{k} \left( \int_{-\infty}^{\infty} \frac{d\theta_i}{2\pi} e^{imx \sinh\Gamma(\theta_i) - imt \cosh\Gamma(\theta_i)} \Gamma'(\theta_i) \right)$$
$$\times \prod_{1 \le i < j \le k} \tanh^2\left( \frac{\Gamma(\theta_i) - \Gamma(\theta_j)}{2} \right), \tag{B.17}$$

where the $\Gamma$ function is given by Eq. (B.15) outside, and by Eq. (B.16) inside the light cone. Note that in the first case, the derivative term simply gives unity, since the contour is just a constant shift.

Now, we are ready to tackle the task of giving a Fredholm determinant representation for Eq. (B.17). The heart of this calculation is the determinant identity

$$\prod_{1 \leq i < j \leq k} \tanh^2\left(\frac{\Gamma(\theta_i) - \Gamma(\theta_j)}{2}\right) = \det\left(\frac{2e^{\Gamma(\theta_i)}}{e^{\Gamma(\theta_i)} + e^{\Gamma(\theta_j)}}\right)^k_{i,j=1} . \tag{B.18}$$

To prove this, we just need to introduce the $u_i = e^{\Gamma(\theta_i)}$ variables and notice that the resulting

$$\prod_{1 \leq i < j \leq k} \left(\frac{u_i - u_j}{u_i + u_j}\right)^2 = \det\left(\frac{2u_i}{u_i + u_j}\right)^k_{i,j=1} , \tag{B.19}$$

expression is trivially true by the famous Cauchy determinant identity

$$\det\left(\frac{1}{x_i + y_j}\right)^k_{i,j=1} = \frac{\prod_{1 \leq i < j \leq k}(x_j - x_i)(y_j - y_i)}{\prod_{1 \leq i,j \leq k}(x_i + y_j)} , \tag{B.20}$$

applied for the special case $x_i = y_i = u_i$. If we combine Eqs. (B.18), (B.17) and (B.7), we arrive at the Fredholm determinant representations

$$C_\pm(x,t) = m^{1/4}\det\left(\mathbb{I} + \mathbf{K}^\pm_{x,t}\right) , \tag{B.21}$$

where the kernels are given by

$$K^\pm_{x,t}(\theta, \theta') = \frac{\pm 1}{2\pi} e^{imx\sinh\Gamma(\theta) - imt\cosh\Gamma(\theta)}\Gamma'(\theta)\frac{2e^{\Gamma(\theta)}}{e^{\Gamma(\theta)} + e^{\Gamma(\theta')}} . \tag{B.22}$$

Finally, using these results, we can express the ferromagnetic and paramagnetic correlation functions in the following way:

$$\begin{aligned}
C_{\mathrm{f}}(x,t) &= \frac{1}{2}m^{1/4}\left(\det\left(\mathbb{I} + \mathbf{K}^+_{x,t}\right) + \det\left(\mathbb{I} + \mathbf{K}^-_{x,t}\right)\right) , \\
C_{\mathrm{p}}(x,t) &= \frac{1}{2}m^{1/4}\left(\det\left(\mathbb{I} + \mathbf{K}^+_{x,t}\right) - \det\left(\mathbb{I} + \mathbf{K}^-_{x,t}\right)\right) .
\end{aligned} \tag{B.23}$$

Equations (B.23) and (B.22) complemented by (B.15) and (B.16) are the most important results of this section regarding the zero temperature case.

The remaining step is to develop analogous results for finite temperatures. First we introduce the finite temperature analogy of Eq. (B.14) using Eq. (22):

$$\begin{aligned}
C_\pm(x,t;\beta) = {}& m^{1/4}S(m\beta)^2 e^{-\Delta\mathcal{E}x}\sum_{k=0}^{\infty}\frac{(\pm 1)^k}{k!}\sum_{\epsilon_1,\ldots,\epsilon_k=\pm}\int_{-\infty+i\epsilon_j\delta}^{\infty+i\epsilon_j\delta}\frac{d\theta_1\ldots d\theta_k}{(2\pi)^k} \\
&\times \prod_{i=1}^{k}\left(\frac{e^{im\epsilon_i(x\sinh\theta_i - t\cosh\theta_i)}}{1 - e^{-\epsilon_i m\beta\cosh\theta_i}}\epsilon_i e^{\epsilon_i\eta(\theta_i)}\right)\prod_{1 \leq i < j \leq k}\tanh\left(\frac{\theta_j - \theta_i}{2}\right)^{2\epsilon_i\epsilon_j} .
\end{aligned} \tag{B.24}$$

Introducing the $u_i = e^{\theta_i}$ variables, the form factor term can be expressed as a determinant:

$$\begin{aligned}
\prod_{1 \leq i < j \leq k}\tanh\left(\frac{\theta_j - \theta_i}{2}\right)^{2\epsilon_i\epsilon_j} &= \prod_{1 \leq i < j \leq k}\left(\frac{u_i - u_j}{u_i + u_j}\right)^{2\epsilon_i\epsilon_j} = \prod_{1 \leq i < j \leq k}\left(\frac{\epsilon_i u_i - \epsilon_j u_j}{\epsilon_i u_i + \epsilon_j u_j}\right)^2 \\
&= \det\left(\frac{2\epsilon_i u_i}{\epsilon_i u_i + \epsilon_j u_j}\right)^k_{i,j=1} = \det\left(\frac{2\epsilon_i e^{\theta_i}}{\epsilon_i e^{\theta_i} + \epsilon_j e^{\theta_j}}\right)^k_{i,j=1} ,
\end{aligned} \tag{B.25}$$

where the third equality holds because it holds for all four $\epsilon_{i,j} = \pm 1$ combinations, and the fourth one is the result of the Cauchy identity given in Eq. (B.20).

Upon substituting Eq. (B.25) into Eq. (B.24) and introducing the new variables $\tilde{\theta}_i = \theta_i - i\epsilon_i\delta$, the correlators become

$$C_\pm(x,t;\beta) = m^{1/4} S(m\beta)^2 e^{-\Delta \mathcal{E} x} \sum_{k=0}^\infty \frac{(\pm 1)^k}{k!} \sum_{\epsilon_1,\dots,\epsilon_k = \pm} \int_{-\infty}^\infty \frac{d\tilde{\theta}_1 \dots d\tilde{\theta}_k}{(2\pi)^k}$$

$$\times \prod_{i=1}^k \left( \frac{e^{im\epsilon_i(x\sinh(\tilde{\theta}_i+i\epsilon_i\delta)-t\cosh(\tilde{\theta}_i+i\epsilon_i\delta))}}{1 - e^{-\epsilon_i m\beta\cosh(\tilde{\theta}_i+i\epsilon_i\delta)}} \epsilon_i e^{\epsilon_i \eta(\tilde{\theta}_i+i\epsilon_i\delta)} \right) \det\left( \frac{2\epsilon_i e^{\tilde{\theta}_i+i\epsilon_i\delta}}{\epsilon_i e^{\tilde{\theta}_i+i\epsilon_i\delta} + \epsilon_j e^{\tilde{\theta}_j+i\epsilon_j\delta}} \right)_{i,j=1}^k . \tag{B.26}$$

Then, using Eq. (B.13), we see that

$$C_\pm(x,t;\beta) = 2\pi C(\beta)^2 e^{\Delta \mathcal{E} x} m^{1/4} \det\left( \mathbb{I} + \tilde{\mathbf{K}}^\pm_{x,t;\beta} \right), \tag{B.27}$$

where the kernels are given by

$$K^\pm_{\epsilon,\epsilon'|x,t;\beta}(\theta,\theta') = \frac{\pm \epsilon e^{im\epsilon(x\sinh(\theta+i\epsilon\delta)-t\cosh(\theta+i\epsilon\delta))+\epsilon\eta(\theta+i\epsilon\delta)}}{2\pi(1 - e^{-\epsilon m\beta\cosh(\theta+i\epsilon\delta)})} \left( \frac{2\epsilon e^{\theta+i\epsilon\delta}}{\epsilon e^{\theta+i\epsilon\delta} + \epsilon' e^{\theta'+i\epsilon'\delta}} \right). \tag{B.28}$$

## C  Algorithm for time-like separations

In this appendix, we present some additional details about the numerical scheme introduced in Section 3.2.1. This discussion will mostly concern optimization techniques useful for reducing computational time. Additionally, we will provide an approximate computational complexity formula for the algorithm.

First, it is advantageous to realize that we are not restricted in using the same $\delta$ rapidity shift for particles and holes. Furthermore, we can make these shifts rapidity-dependent as long as kinematic poles are disallowed. We should change our perspective to exploit the full potential coming from these additional degrees of freedom. Let us fix the real and imaginary parts of the ray parameters,

$$\begin{aligned} -\text{Im}\,\zeta_+ = \text{Im}\,\zeta_- &\equiv \text{Im}\,\zeta, \\ \text{Re}\,\zeta_+ = \text{Re}\,\zeta_- &\equiv \text{Re}\,\zeta, \end{aligned} \tag{C.1}$$

and then adjust the contours accordingly.

The conditions in Eqs. (36) are reformulated as

$$\begin{aligned} \delta_+ &< \delta_{c+} = \arctan\left( \frac{\text{Im}\,\zeta}{\text{Re}\,\zeta - 1} \right), \\ \delta_- &< \delta_{c-} = \arctan\left( \frac{\text{Im}\,\zeta + \beta/x}{\text{Re}\,\zeta - 1} \right), \end{aligned} \tag{C.2}$$

where $\delta_\pm$ are the rapidity shifts for particles and holes. However, these constraints come from large positive rapidities: negative rapidities do not cause any singular behavior in Eq. (36a) or Eq. (36b). Therefore, we can incorporate an appropriate rapidity-dependence in Eq. (C.2) to loosen the constraints. For the particles, an appropriate choice is

$$\delta_+(\theta) = \frac{\pi}{8}(1 - \tanh\theta). \tag{C.3}$$

This function approaches zero for large rapidities, so it satisfies Eq. (C.2) on the relevant domain while becoming approximately $\pi/4$ for small rapidities and hence reducing the oscillatory behavior there. We could use a similar contour prescription for holes but for any

reasonable choice of $m\beta, x$ and $\text{Re}\,\zeta$, the thermal part of Eq. (C.2) regularizes the expression such that the oscillatory behavior does not pose any serious issue. Therefore, we used

$$\delta_- = \frac{\delta_{c-}}{2}\,, \tag{C.4}$$

as the hole rapidity shifts. The factor of 2 is a numerical choice that proved sufficient for all practical purposes.

Next, it is important to discuss the relevant rapidity range $\theta \in [\vartheta_1, \vartheta_2]$. As stipulated by the condition in Eq. (C.2), the integrand of Eq. (35) vanishes exponentially for large (positive or negative) rapidities. This exponential cutoff is governed by the

$$f_\epsilon(\theta) = \frac{e^{imx\epsilon(\sinh(\theta+i\epsilon\delta_\epsilon(\theta))-(\text{Re}\,\zeta-i\epsilon\text{Im}\,\zeta)\cosh(\theta+i\epsilon\delta_\epsilon(\theta)))}}{1 - e^{-\epsilon m\beta\cosh(\theta+i\epsilon\delta_\epsilon(\theta))}} \tag{C.5}$$

envelope. Then, we can introduce a parameter $A > 0$ characterizing the accuracy of the computations and calculate the relevant rapidity range by numerically solving the

$$|f_+(\theta)| = e^{-A}, \qquad |f_-(\theta)| = e^{-A} \tag{C.6}$$

equations. These have two solutions corresponding to $L_{1\pm}$ and $L_{2\pm}$, respectively, so

$$\begin{aligned} L_1 &= \min\{L_{1+}, L_{1-}\}\,, \\ L_2 &= \max\{L_{2+}, L_{2-}\}\,. \end{aligned} \tag{C.7}$$

For most calculations, we used the $A = 10$ value. Note that for a wide range of parameter choices, the value of $L_2$ can be approximated by

$$L_2 \approx \log\left[\frac{2A}{mx\text{Im}\,\zeta}\right]\,, \tag{C.8}$$

which will turn out to be useful for estimating the computational complexity.

The last step is to analyze the rapidity discretization. For simplicity, we consider a constant $\Delta\theta$ step size between sampling points, but the approach can easily be generalized to an adaptive method. For generic parameter settings, the thermal part of $\delta_{c-}$ in Eqs (C.2) and (C.4) are large enough such that kinematic poles do not pose any problem. Therefore the oscillatory behavior of the integrands gives the relevant criterion for the discretization. Due to the aforementioned thermal effects and the contour prescription of Eq. (C.3), only the large rapidity behavior of the particle contributions is significant. Therefore, we can use the rapidity $L_2$ to set the condition

$$|g'(L_2)|\Delta\theta = 1\,, \tag{C.9}$$

where $g(\theta)$ is the relevant oscillatory factor:

$$g(\theta) = imx\left(\sinh[\theta + i\delta_+(\theta)] - (\text{Re}\,\zeta - i\text{Im}\,\zeta)\cosh[\theta + i\delta_+(\theta)]\right)\,. \tag{C.10}$$

A numerical solution of this equation completes the algorithm: after these considerations, the simulations are completely analogous to the space-like case.

We can approximate the solution of Eq. (C.9). For a generic parameter choice $\delta_+(L_2) \approx 0$, so from Eq. (C.8) we obtain

$$\Delta\theta \approx \frac{\text{Im}\,\zeta}{A(\text{Re}\,\zeta - 1)}\,. \tag{C.11}$$

Therefore, we can make a rough estimate of the number of rapidity points we need:

$$N \approx L_2/\Delta\theta = \frac{A(\text{Re}\,\zeta - 1)}{\text{Im}\,\zeta}\log\left[\frac{2A}{mx\text{Im}\,\zeta}\right]\,. \tag{C.12}$$

The most important message of this formula is that $N \sim (\text{Re}\,\zeta - 1)$. Since calculating the determinant of the discretized matrix takes $O(N^3)$ steps, it is unfeasible to perform simulations for large $\zeta$ values.

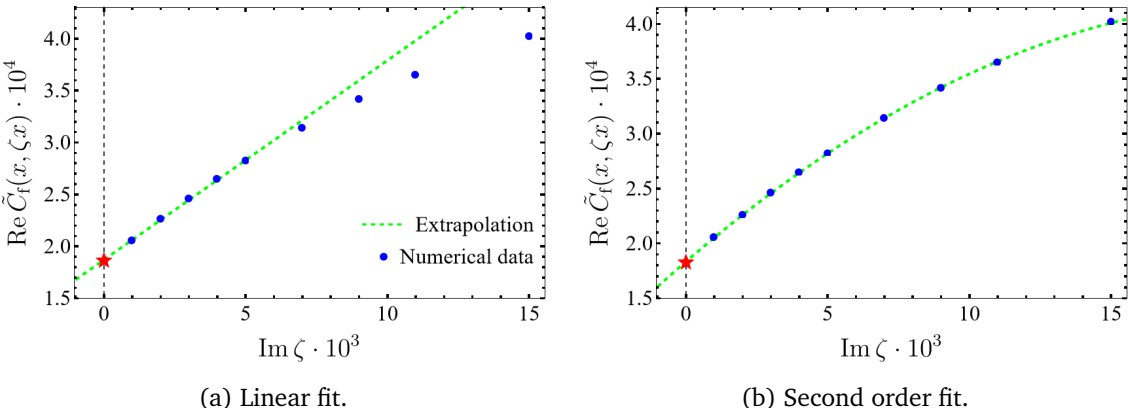

(a) Linear fit.  (b) Second order fit.

Figure 15: Extrapolation scheme for calculating the physical correlation functions corresponding to $\mathrm{Im}\,\zeta = 0$ (red stars). The blue dots represent simulation data for $mx = 19.5$, $\mathrm{Re}\,\zeta = 1.5$, and $m\beta = 1$; the green dashed lines are the extrapolating functions. On the left, a linear fit was used for the smallest five $\mathrm{Im}\,\zeta$ values, while on the right all points are part of a quadratic fit.

## C.1 Extrapolation scheme

As discussed in Sec. 3.2.1, the last part of the time-like algorithm is an extrapolation procedure by which we estimate the results at the physical $\mathrm{Re}\,\zeta_+ = \mathrm{Re}\,\zeta_-$, $\mathrm{Im}\,\zeta_+ = \mathrm{Im}\,\zeta_- = 0$ point.

Here we show an example for this extrapolation procedure at $m\beta = 1$ in Fig. 15. Fixing $\mathrm{Re}\,\zeta_+ = \mathrm{Re}\,\zeta_- = 1.5$, we used the same imaginary ray parameter $\mathrm{Im}\,\zeta$ for particles and holes and investigated the convergence properties as $\mathrm{Im}\,\zeta \to 0$. For all parameter settings, we found that sufficiently close to the physical point,

$$C_{p,f}(x,\zeta_+,\zeta_-,\beta) \approx C_{p,f}(x,\zeta,\beta) + c_1 \mathrm{Im}\,\zeta + O(\mathrm{Im}\,\zeta^2), \tag{C.13}$$

where $c_1$ is a (complex) numerical constant. Therefore we could obtain the physical correlators by fitting a line on the $C_{p,f}(\zeta)$ vs $\mathrm{Im}\,\zeta$ curve and reading off the $y$ intercept. This process is presented in Fig. 15a. However, decreasing the imaginary part of $\zeta$ is computationally costly: for certain parameter settings, especially for large $mx$ values, it was not possible to reach the applicability range of Eq. (C.13). In these cases, we used a quadratic fit,

$$C_{p,f}(x,\zeta_+,\zeta_-,\beta) \approx C_{p,f}(x,\zeta,\beta) + c_1 \mathrm{Im}\,\zeta + c_2 \mathrm{Im}\,\zeta^2 + O(\mathrm{Im}\,\zeta^3), \tag{C.14}$$

as shown in Fig. 15b. This protocol proved to be sufficient in all cases.

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
