# Peer review of "Dynamical correlation functions in the Ising field theory"

_SciPost Physics, doi:SciPost Phys. 17, 162 (2024)_

## Round 2 · Referee Report · Dirk Schuricht (Referee 1) · 2024-7-21

Report

The article studies the two-point correlation functions of the magnetisation in the Ising field theory at finite temperature. The authors perform a finite-temperature form factor expansion and provide a resummation of it in terms of Fredholm determinants, which can be viewed as a major achievement. This allows a detailed analysis, in particular a numerical evaluation. The obtained results are in perfect agreement with previous results in certain special and limiting cases. The manuscript very clearly presents the derivation and results. Thus I recommend publication in SciPost Physics.

I have two minor questions/remarks, that the authors should consider: 1. At short distances the results from the numerically evaluated Fredholm determinant are compared to conformal field theory predictions. I wonder whether it is possible to obtain the resulting scaling behaviour, like (4.1), analytically from the determinant. 2. The non-analyticity of the correlation length (5.14) shown in Fig. 5.8 originates from the switch of the leading exponential term in the expansion (5.11). Is it feasible to study the effect of the sub-leading term, ie, the crossing between the two leading exponentials in more detail?

Recommendation

Publish (meets expectations and criteria for this Journal)

  • validity: -
  • significance: -
  • originality: -
  • clarity: -
  • formatting: -
  • grammar: -

Author:  István Csépányi  on 2024-08-20  [id 4705]

(in reply to Report 1 by Dirk Schuricht on 2024-07-21)
Category:
answer to question

We thank the Referee for his careful reading of our manuscript and we are glad that he recommends the publication of our paper in SciPost Physics. Below we provide our answers to the insightful questions raised by the Referee:

  1. This is a very interesting question. To our knowledge, the Fredholm determinant representation can be used to derive the long distance (or time) asymptotic behavior. We do not know if it is possible to obtain the short distance behavior but it would certainly be interesting. However, we refrained from such analytic derivations as they are beyond the scope of our paper. The analytic derivation of both the large and short distance asymptotics directly from the Fredholm determinant representations is an exciting line of future research.

  2. Thank you for this thoughtful question. The non-analyticity indeed stems from the switch between the leading exponential terms. However, it only arises when taking the limit x→ $\infty$ asymptotic limit, so finding a clear crossover behavior between the two leading exponentials in a finite regime is challenging. The correlators themselves are smooth functions of the ray parameter. Regardeless, one could investigate whether including more sub-leading terms in the expansion improves the approximation and smoothens the transition. To explore this, we performed additional simulations at $m\beta = 5$ and $\zeta = 0.75$, where the leading term corresponds to $k_0 = 1$ (see Eq. (5.13)). The attached plot shows the simulated data, the leading decay corresponding to this $k = 1$ value, and a refinement including the $k = 0$ and $k = 2$ terms as well. We can clearly see that keeping more terms makes the approximation better for smaller mx values, which might be phrased as a crossover in the spatial separation.

Attachment:

---

## Round 2 · Referee Report · Anonymous (Referee 2) · 2024-10-16

Strengths

  • Thorough technical analysis of the dynamical correlations
  • Generality of the method
  • Unexpected outcome
  • Thoughtful comparison with known results

Weaknesses

  • The asymptotic analysis is mainly numerical
  • The paper is technical

Report

The authors study the non-equal time correlation functions of the longitudinal magnetization in the 1D Ising field theory at finite (and zero) temperature. The strongest point of the investigation is that it provides a method to compute the correlation functions in all space-time and temperature regimes both in the ferromagnetic and in the paramagnetic phase. The method is very interesting and the outcomes of the investigation include a quite unexpected result: in the paramagnetic phase, the correlation length associated with the two-point function at space-like distances is not continuously differentiable with respect to both the temperature and the ballistic-scale coordinate $\zeta=t/x$. The presentation is good and the messages are clearly stated. Thus, I strongly recommend this paper for publication.

Some minor comments follow: 1- I don’t like the hybrid notations in (2.6); I suggest the authors to consider the possibility to give up the use of sums over momenta in the scaling limit, where the use of integrals is more conventional. 2- Does the non-analytic behaviour of the correlation length (in the paramagnetic phase) implies that the the non-equal time correlation between the magnetization in two orthogonal directions (which is the time derivative of the non-equal time correlation of the longitudinal field) is a discontinuous function of the temperature? If so, what would happen by taking a second time derivative? 3- Is it correct to interpret that non-analitic behaviour as an effect of crossovers of eigenvalues of some transfer operator associated with the correlation function? 4- In other works, the scaling variable $\zeta$ denotes the inverse of what is used in this paper, i.e., $x/t$.

Requested changes

  • I do not request specific changes.

Recommendation

Publish (surpasses expectations and criteria for this Journal; among top 10%)

  • validity: top
  • significance: high
  • originality: high
  • clarity: high
  • formatting: excellent
  • grammar: excellent

Author:  István Csépányi  on 2024-10-23  [id 4899]

(in reply to Report 2 on 2024-10-16)
Category:
answer to question

We thank the Referee for the thorough reading of our manuscript. We are pleased by their recommendation for the publication of our paper in SciPost Physics. The Referee made some interesting comments which we answer below.

1- For the sake of clarity, we decided to take the scaling limit while keeping the volume of the system, $L=Na$, fixed. As a result, the momenta are still quantised even in the field theory. However, in the rest of the paper, we indeed work in the thermodynamic limit, where such sums become integrals. The Referee's comment made us realise that this was not discussed clearly, so we have added a sentence to the beginning of the paragraph below Eq. (2.7).

2- This is an interesting question. The non-analytic behaviour in the form of cusps appears in the correlation length as a function of the temperature or of $\zeta=t/x$. Taking time derivatives does not modify the non-analytic behaviour in temperature but one may wonder what happens with the $\zeta$ dependence. Importantly, the correlation length describes the asymptotic behaviour in the $x\to\infty$ limit taken at a fixed value of $\zeta$. The time derivative of the function giving this asymptotic behaviour does not give in general the asymptotic behaviour of the time derivative of the correlation function. Thus the correlation function itself can be, and we believe is, continuous in $x$ and $t$.

3- We are not aware of transfer matrix approaches that would yield the asymptotic behaviour for fixed $\zeta$, let alone in the field theory. But it is of course possible that such an approach exists or will be constructed in the future. In this case, it is quite possible that the cusps we found correspond to crossing eigenvalues of a transfer matrix.

4- The notation $\zeta=x/t$ is probably more common, however, we used $\zeta=t/x$ as it was more natural and convenient because we discussed the equal time case ($\zeta=0$) but not the autocorrelation function ($\zeta=\infty$).

---

## Editorial Decision

published